# Antagonism between killer yeast strains as an experimental model for biological nucleation dynamics

Andrea Giometto[1,2,3]*, David R Nelson[2,3,4], Andrew W Murray[3]

[1]School of Civil and Environmental Engineering, Cornell University, Ithaca, United States; [2]Department of Physics, Harvard University, Cambridge, United States; [3]Department of Molecular and Cellular Biology, Harvard University, Cambridge, United States; [4]John A Paulson School of Engineering and Applied Sciences, Harvard University, Cambridge, United States

**Abstract** Antagonistic interactions are widespread in the microbial world and affect microbial evolutionary dynamics. Natural microbial communities often display spatial structure, which affects biological interactions, but much of what we know about microbial antagonism comes from laboratory studies of well-mixed communities. To overcome this limitation, we manipulated two killer strains of the budding yeast *Saccharomyces cerevisiae*, expressing different toxins, to independently control the rate at which they released their toxins. We developed mathematical models that predict the experimental dynamics of competition between toxin-producing strains in both well-mixed and spatially structured populations. In both situations, we experimentally verified theory's prediction that a stronger antagonist can invade a weaker one only if the initial invading population exceeds a critical frequency or size. Finally, we found that toxin-resistant cells and weaker killers arose in spatially structured competitions between toxin-producing strains, suggesting that adaptive evolution can affect the outcome of microbial antagonism in spatial settings.

*For correspondence: giometto@cornell.edu

**Competing interest:** The authors declare that no competing interests exist.

## Introduction

Microbes affect nearly every aspect of life on Earth, from carbon fixation (*Falkowski et al., 1998*) to human health (*Srivastava and Bhargava, 2016*). They often live in dense aggregates, such as biofilms, which offer them protection from environmental forces, drugs, and predation (*Nadell et al., 2016*). To prosper in these dense communities and to resist external attacks or takeover from cheater phenotypes, microbes display a wide range of social interactions (*West et al., 2007*), both cooperative, such as cross-feeding and quorum sensing, and antagonistic, such as toxin and antibiotic production. The high densities and close proximity of the members of cellular aggregates affect these social interactions, which in turn affect the spatial structure and the spatiotemporal dynamics of microbial communities (*Kayser et al., 2018a*).

Laboratory experiments with genetically engineered microbes have helped us understand how social interactions can alter the evolutionary dynamics of microbial populations (*Amor et al., 2017*; *McNally et al., 2017*; *Müller et al., 2014*; *Celik Ozgen et al., 2018*; *Weber et al., 2014*). For example, cooperation, in which two strains feed each other amino acids, prevents the separation between different genotypes (*Müller et al., 2014*) that occurs when two non-interacting populations spread across a surface in a range expansion (*Hallatschek et al., 2007*). Antagonistic interactions are found in both prokaryotes (*Atanasova et al., 2013*; *Cheung et al., 1997*; *Riley and Wertz, 2002*; *Veening and Blokesch, 2017*) and eukaryotes (*Boynton, 2019*). They occur in many ecological niches such as the rhizosphere (*Kent and Triplett, 2002*), aquatic systems (*Feichtmayer et al., 2017*; *Drebes*

*Dörr and Blokesch, 2020*), and human infections (*Schoustra et al., 2012*; *Libberton et al., 2015*; *Heilbronner et al., 2021*), and are frequently exploited for biocontrol applications (*Kim et al., 2006*; *Weller, 2007*). These interactions are typically mediated by toxins that are produced and released by cells, for example, bacteriocins (*Schoustra et al., 2012*) or antibiotics (*Granato et al., 2019*), or injected directly into neighboring cells, as in the case of type VI secretion systems (*Borgeaud et al., 2015*; *Granato et al., 2019*). On an agar plate, the ability of two *Vibrio cholerae* strains to kill each other, using the type VI secretion system, coarsens the single-strain domains of populations that were initially well mixed (*McNally et al., 2017*; *Yanni et al., 2019*).

Theoretical models predict different outcomes for cooperation and antagonism: cooperators require each other to prosper (*Müller et al., 2014*) and antagonistic interactions lead to the competitive exclusion of one of the antagonists (*Lavrentovich and Nelson, 2019*; *Nowak et al., 2004*; *Tanaka et al., 2017*). We refer to the strain that survives in a 1:1, well-mixed culture as the stronger antagonist and the one that goes extinct as the weaker antagonist. Models based on generalizations of the Lotka-Volterra equations (*Lavrentovich and Nelson, 2019*; *Tanaka et al., 2017*) predict that being a stronger antagonist is a necessary, but not a sufficient condition for an invading strain to replace a resident, antagonist population: successful replacement requires that the initial inoculum of the invading antagonist be larger than a critical frequency (i.e., relative abundance) in well-mixed populations or a critical size in spatially structured populations. For simplicity, we refer to critical frequency in well-mixed populations and critical size in spatially structured ones as 'critical inoculum size'. The prediction of a critical size has implications for the population dynamics of antagonistic interactions: the requirement for a critical inoculum size implies that mutations conferring an increased strength of antagonism may not necessarily establish in a population because of a deterministic push to extinction if the population size of the mutant is below a given threshold. Conversely, the critical size predicts that populations of weaker antagonists should be resistant to invasion from a stronger antagonist, at least below a certain rate of immigration. Finally, a critical inoculum size has implications for the possibility of exploiting antagonistic, microbial interactions to manipulate microbiomes: exploiting antagonistic interactions may allow us to design microbial consortia that are resistant to external invasion, but if we want to manipulate natural communities, engineered strains will need to be introduced into the microbial community at sufficiently large densities (*de Lorenzo et al., 2016*). The prediction that being a stronger antagonist is necessary, but not sufficient, to invade a resident population requires experimental verification, which motivated our work.

We developed an experimental system in which two strains of the budding yeast, *Saccharomyces cerevisiae*, expressed two different toxins from two different, inducible promoters (*Figure 1A*). We

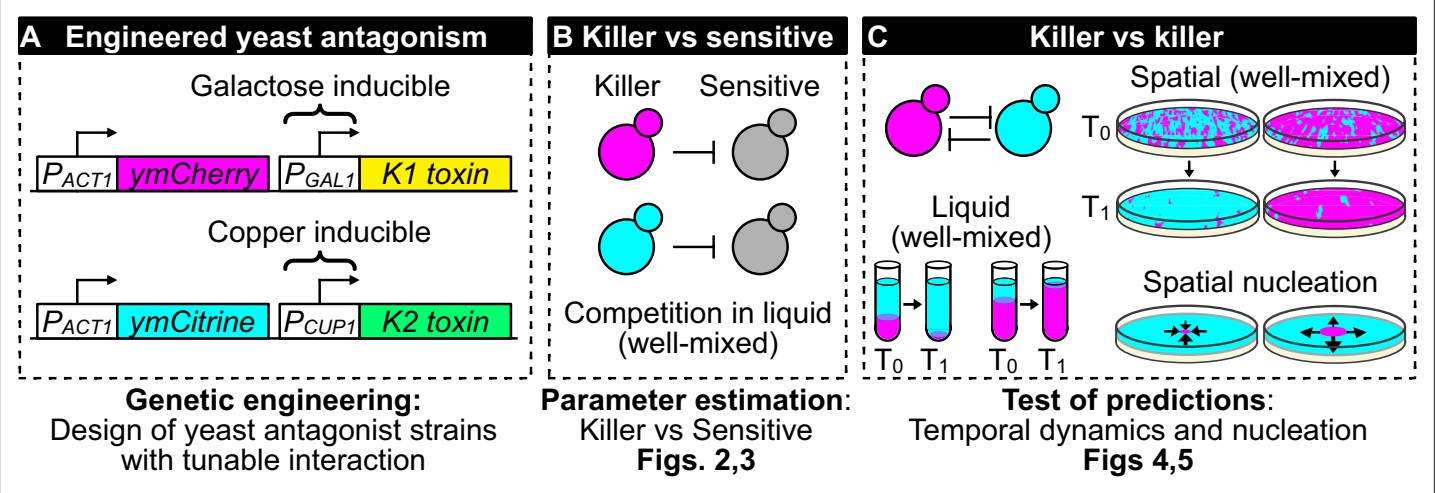

**Figure 1.** Overview of genetic engineering and experiments performed to investigate the population dynamics of microbial antagonism. (**A**) We genetically engineered yeast strains to express two different fluorescent proteins (ymCherry and ymCitrine) constitutively and two different toxin/immunity genes (K1 and K2) in response to the inducers galactose (via the $P_{GAL1}$ promoter) and copper (via the $P_{CUP1}$ promoter). (**B**) We used competition assays between toxin-producing cells ('killer cells' in cyan and magenta) and sensitive, nonkiller cells (gray cells) to parametrize mathematical models of toxin production, cell growth, and toxin-induced cell death. (**C**) We used the models and the experimental system to investigate population dynamics in the presence of antagonistic interactions in both well-mixed (in liquid and on surfaces) and spatially structured populations on surfaces.

investigated the dynamics of competitive exclusion in three environments: spatially well-mixed populations in liquid cultures or on surfaces, and spatially structured populations on surfaces. We derived mathematical models of population dynamics regulated by toxin production and toxin-induced cell death and parametrized them using competition assays between toxin-producing ('killer') cells and sensitive, nonkiller ones (*Figure 1B*). Experiments verified theoretical predictions on the conditions that lead to a successful invasion of an antagonistic strain in all three environments (*Figure 1C*). The mathematical models correctly predict the dynamics of competition between toxin-producing strains in all scenarios considered here, they highlight the processes that lead to a region devoid of cells between two antagonistic strains that encounter each other on a solid surface, and can guide attempts to manipulate naturally occurring microbial communities.

We begin by discussing the experimental system (*Figure 1A*) and the parametrization of mathematical models of antagonism using well-mixed experiments (*Figure 1B*). Then, we verify the predictions of these models for the competition of two mutually antagonistic strains in three settings: well-mixed cultures in liquid, well-mixed communities on surfaces, and spatially structured communities on surfaces (*Figure 1C*). We then discuss mathematical models that give us intuition for the formation of depletion zones at the interface between two antagonist strains in spatially structured communities and finally we discuss mutants that appeared during the experiments and affected the dynamics of antagonism.

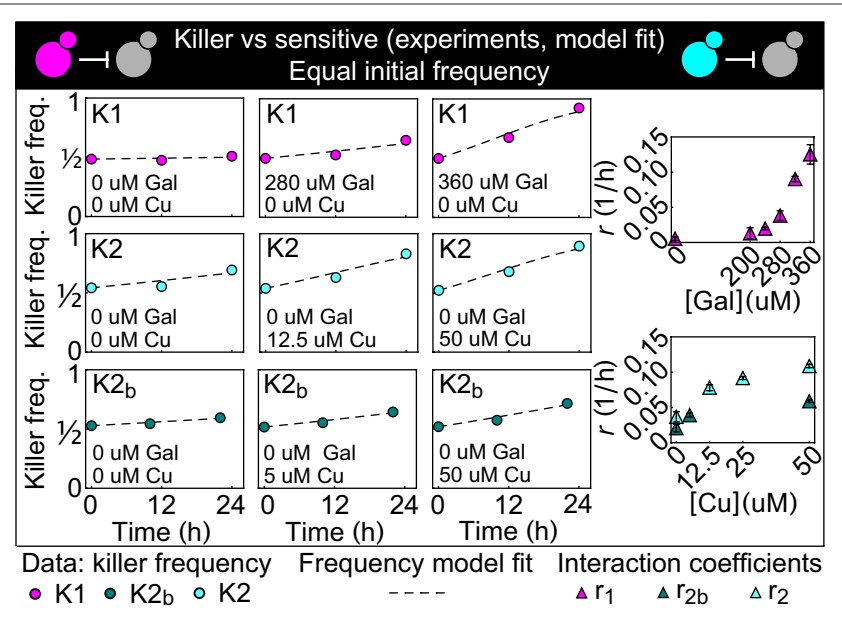

**Figure 2.** Temporal change of the killer strain frequencies in competition assays against sensitive strains, at different inducer concentrations. Different colored points depict data corresponding to the killer strains K1 (magenta), K2 (cyan), and K2$_b$ (dark green). For each killer strain, increasing its inducer's concentration increased its killing strength, that is, the rate at which its frequency grew with time. In the absence of inducers, the K1 strain frequency remained constant, whereas strains K2 and K2$_b$ still displayed killing activity, which we attribute to the leakiness of the $P_{CUP1}$ promoter. Each data point is the mean of two, three, or five technical replicates. The x axis reports time since the first measurement. Dashed lines show the best fits of the frequency model *Equation 1* and *10*. The panels on the right show the best-fit interaction coefficients (which are proportional to toxin production rates) as a function of the inducer concentrations (mean ± SD, Table 3).

The online version of this article includes the following source data for figure 2:

**Source data 1.** This Excel spreadsheet contains all data used to plot *Figure 2* and to compute the interaction coefficients.

## Results

### Competition between killer and sensitive strains measures toxin production

We began by using competition between killer and nonkiller strains in liquid cultures to estimate the parameters needed to model the antagonism between two different killer strains, and to test our ability to vary the strength of the interaction by changing the concentration of the two inducers. The two killer strains expressed two different killer proteins (*Tipper and Bostian, 1984*) that do not confer immunity to each other: strain K1 expresses the killer toxin K1 (*Bevan and Makower, 1963*) and strain K2 expresses the killer toxin K2 (*Naumova and Naumova, 1973*). Killer cells are immune to the toxin they produce because the unprocessed toxins confer immunity (*Dignard et al., 1991*; *Hanes et al., 1986*). Both the K1 and K2 killer toxins bind to β-1,6-glucans on the cell wall, and subsequently translocate to the cytoplasmic membrane where they bind to a secondary receptor (Kre1p for K1, an unknown receptor for K2). Both K1 and K2 disrupt the cytoplasmic membrane increasing its permeability to ions (*Magliani et al., 1997*). The nonkiller strains S1 and S2 carried genetic constructs like those in the K1 and K2 strains, but without the killer toxin genes. As a result, S1 expresses the same fluorescent proteins as K1 and S2 expresses the same fluorescent protein as K2, allowing us to distinguish killer cells from nonkiller ones (competing K1 against S2 and K2 against S1), and thus to measure the cell densities of the two strains with a flow cytometer. *Figure 2* shows the result of mixing killer and sensitive strains in a 1:1 ratio and following the fraction of the killer strain against time. In the absence of the inducer, galactose, the frequency of K1 remained constant. Increasing the concentration of galactose increased the rate at which the frequency of K1 grew with time. For the K2 toxin, we used two killer strains which had the same genetic construct integrated into their genome, but displayed different fluorescent intensities and different killing strengths (i.e., the rates at which they increased their frequency with time), revealing that the genetic construct was integrated at different copy numbers in the two strains, K2 and $K2_b$. For both strains, their relative frequency increased with time even in the absence of copper, which is consistent with leaky expression from the $P_{CUP1}$ promoter (*Butt et al., 1984*; *Gorman et al., 1986*) in the absence of added copper. Increasing the concentration of copper increased the rate at which the two strains' frequencies grew with time; the less fluorescent strain $K2_b$ was a weaker killer than the more fluorescent strain K2 at all copper concentrations.

We developed a simple mathematical model of cell growth, toxin production, and toxin-induced cell death, and used the data in *Figure 2* to fix the parameters. Under suitable assumptions on the relative time scales of cell division and toxin production (Materials and methods), the temporal change of the K1's frequency, $f$, in competition against a K2 killer strain under well-mixed conditions can be described by a single equation:

$$\frac{df}{dt} = f\left(1 - f\right)\left[r_1 f - r_2\left(1 - f\right)\right], \tag{1}$$

where $df/dt$ denotes the temporal derivative of $f$, and $r_1$ and $r_2$ are interaction coefficients proportional to the toxin production rates of strains K1 and K2 (see Materials and methods). For two strains K1 and K2 at equal initial frequencies (i.e., $f_0 = 1/2$), a Taylor expansion of the solution to *Equation 1* around $f_0$ shows that, initially, $f$ varies linearly with time with a rate proportional to $r_1 - r_2$, that is, $f(t) = (r_1 - r_2)\, t/8 - O\left(f - 1/2\right)$, before non-linear terms become important. For a killer strain competing against a sensitive, nonkiller one, *Equation 1* reduces to $df/dt = rf^2\left(1 - f\right)$, where $f$ is the killer strain frequency. The best fits of this last equation are shown as dashed lines in *Figure 2*, and the best-fit estimates of the interaction coefficient $r$ for the three strains K1, K2, and $K2_b$ at different inducer concentrations are shown in the lower panels (numerical values are given in Table 3). A formally equivalent model that included spatial diffusion and noise due to number fluctuations was studied theoretically in *Lavrentovich and Nelson, 2019*, where it was derived starting from a stepping-stone model with local, antagonistic interactions, that is, without explicitly modeling the secretion of diffusible toxins. When toxin dynamics are much faster than cell density dynamics, our model and the well-mixed version of the earlier model (*Lavrentovich and Nelson, 2019*) coincide (Materials and methods).

According to *Equation 1*, the dynamical system describing the antagonistic interaction of two killer strains has two stable equilibria, one at $f = 0$ and one at $f = 1$, and one unstable equilibrium at $f_{eq} = r_2/\left(r_1 + r_2\right)$. If the initial frequency is above $f_{eq}$ the system tends to $f = 1$, otherwise it tends

to $f = 0$. In other words, the strain K1 can only increase its frequency in the population if its initial frequency is larger than the critical inoculum frequency, $f_{eq}$. The equilibrium frequency $f_{eq}$ thus represents a critical inoculum size below which the invasion of a stronger antagonist is predicted to fail in well-mixed settings. Note that this particular 'size' relates to an inoculum *concentration* rather than the actual physical size discussed later in this paper for spatially structured communities on surfaces. Nevertheless, when number fluctuations are included in the dynamics, there is an interesting analogy with escape over a barrier problems in statistical mechanics (*Chotibut and Nelson, 2015*). Increasing the toxin production rate of strain 1 increases $r_1$ and is thus predicted to decrease the size of the critical inoculum. An intuitive derivation for the critical frequency $f_{eq}$ can be obtained by assuming that the two toxins have equal per-cell binding rates and kill cells of the other strain at the same rate: With this assumption, $r_1$ and $r_2$ are proportional to the per-cell toxin production rate of the strains K1 and K2, and the equilibrium frequency $f_{eq}$ is the frequency at which the populations of the two strains produce the same amount of toxins per unit time. In the general case in which the two toxins have different per-cell binding and killing rates, the equilibrium frequency $f_{eq}$ is such that the toxin-induced, per-capita death rates for the two strains are equal. A critical inoculum size is thus present in this system because a stronger antagonist must overcome the toxin production from its competitor, before being able to expand in the population. *Figure 3C* plots the rate at which the fraction of strain 1 changes at different interaction coefficients, which are determined by the concentrations of galactose and copper, the inducers of toxin production. *Equation 1* can also be rewritten as $df/dt = -dV/df$, Done.where $V$ is the quartic potential depicted in *Figure 3C*. When both $r_1$ and $r_2$ are positive, the potential $V$ has a double-well structure with two minima, corresponding to the two stable states in which one strain competitively excludes the other. Separating the two minima is an energy barrier having its peak at the critical frequency that the first strain must overcome to exclude the second. As shown theoretically in *Lavrentovich and Nelson, 2019*, the spatial, stochastic generalization of *Equation 1* can be interpreted as an escape over the barrier problem, and lends itself to the use of theoretical techniques from nucleation theory as first appreciated by *Rouhani and Barton, 1987*, in the context of spatial population genetics.

## A simple model predicts the competition between two antagonistic killer strains

Having measured the two interaction coefficients ($r_1$ and $r_2$) in competitions involving antagonism acting on sensitive strains, we asked if our model could predict the frequency dynamics of the two killer strains K1 and K2 competing against each other. *Figure 3A* shows the frequencies of the K1 and K2 strains grown in liquid following the same protocol as the experiments of *Figure 2*, starting from equal frequencies for the two strains and varying the concentrations of the inducers, which control $r_1$ and $r_2$. The frequency of K1 increased if $r_1 > r_2$ and decreased otherwise, in accordance with *Equation 1* above. *Figure 3B* shows the frequencies of the two strains separated by an interval of 24 hr, with the frequencies at 14 hr after inoculation on the x axis and the frequencies at 38 hr after inoculation on the y axis. The insets in panel B show two control experiments consisting of competition assays between the nonkiller strains S1 and S2: the two strains have identical fitness, so their relative frequency remains constant over 24 hr of growth. The model predictions (solid line) and the 68% confidence intervals (gray shading) reveal that the model can predict the temporal dynamics and the value of the unstable equilibrium $f_{eq}$ (last panel in *Figure 3B*), for all inducer concentrations and for all initial frequencies. The ability of parameters estimated from killer versus sensitive assays (*Figure 2*) to predict the dynamics of the competition between two killer strains shows that the interaction terms included in *Equation 1* are sufficient to capture the antagonistic dynamics: we can simply sum the contribution of strain K2 to the death rate of strain K1 (the term $r_2 (1 - f)$ in *Equation 1*, see Materials and methods) and the corresponding contribution of strain K1 to the death rate of strain K2 (the term $r_1 f$) without adding additional terms to the equation. The experiments also show that for initial frequencies close to the unstable equilibrium, different technical replicates can tend toward different stable equilibria in the long-term limit (*Figure 3D*), highlighting the instability of the equilibrium $f_{eq}$ and the fact that the energy barrier can be overcome when the initial frequency is close to $f_{eq}$. *Figure 3—figure supplement 2* shows that there is no correlation between the frequency at the first and second measurement time point for those replicates that were initialized close to the unstable equilibrium (i.e., data points at 250 μM galactose in *Figure 3D*), suggesting that the stochasticity of the dynamics dominates over

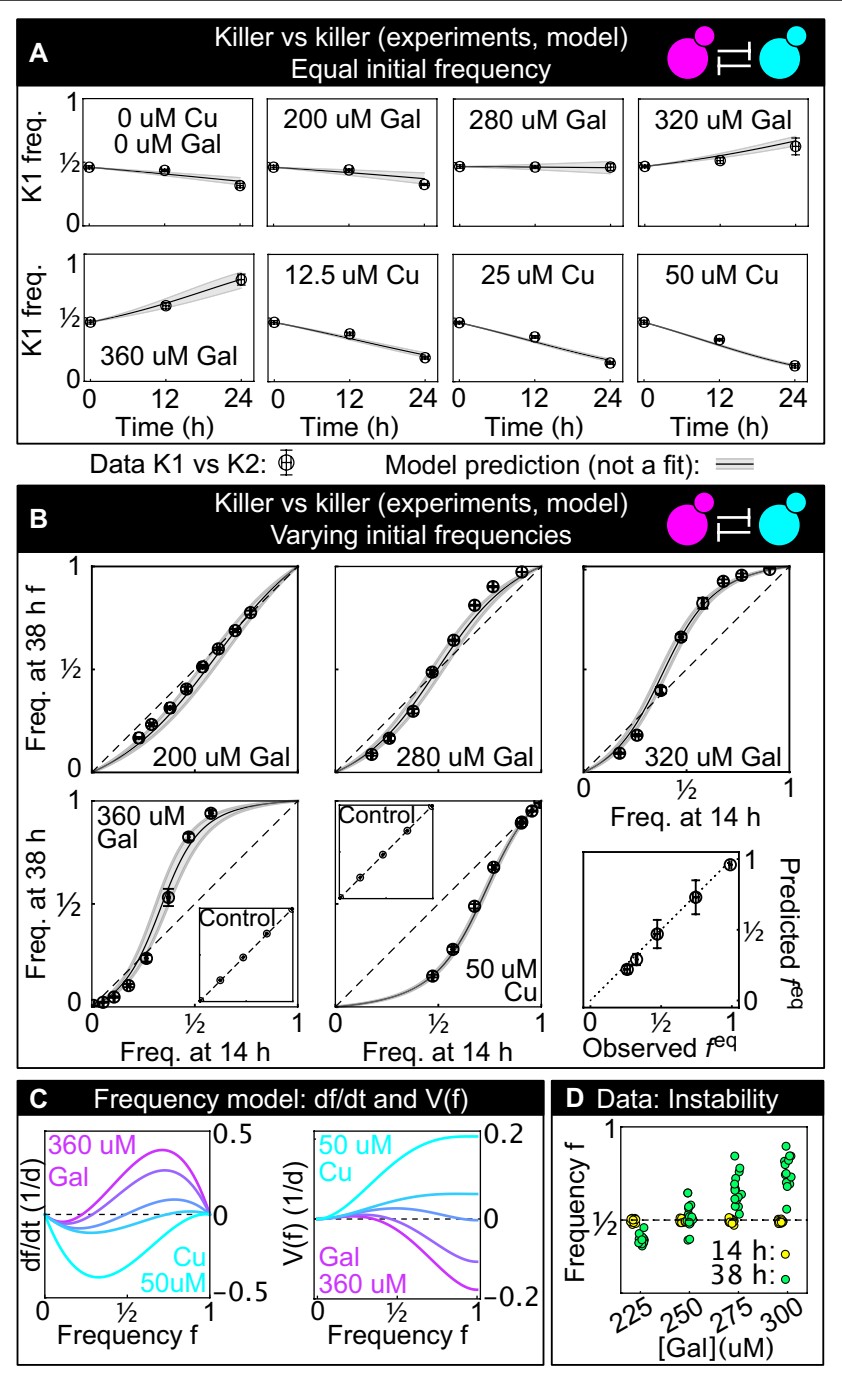

**Figure 3.** Comparison between experimental, well-mixed competitions of killer vs killer and model predictions. (**A**) Temporal change of strain K1's frequency in competition assays against strain K2, at different inducer concentrations (only one of the two inducers was added in each replicate). Each data point is the mean of 13 or more technical replicates, error bars are one standard deviation. Solid lines in (**A**) and (**B**) are predictions according to *Equation 1* with parameters estimated from competitions between the killer strains K1 and K2 and the sensitive, nonkiller strains S1 and S2. Gray bands show the 68% confidence interval for the model. (**B**) Changes in the frequency of strain K1 following 24 hr of competition against strain K2, at different concentrations of the inducers (subpanels) and at different initial frequencies. The x axis gives the K1 frequency 14 hr after inoculation, the y axis gives the K1 frequency 38 hr after inoculation. The dashed lines show the 1:1 line that points would lie on if the two strains had equal fitness. The critical inoculum corresponds to the intersection point between the dashed lines and the solid lines (model), or an interpolation of the data points (experiment). Insets show the control experiments

*Figure 3 continued on next page*

*Figure 3 continued*

of competing the two sensitive strains S1 and S2 with each other at the same inducer concentrations as the parent subpanels. The bottom-right subpanel shows the correlation between the value of $f_{eq}$ predicted from our model (the intersection point between the solid and dashed lines) and the experimental value of $f_{eq}$ based on competitions (the intersection of the interpolation between the data points and the dashed lines) for all inducer concentrations. The dotted line is the 1:1 line. (**C**) Temporal derivative of $f$ according to **Equation 1**, at the inducer concentration values of panel B (left, only the lines corresponding to the highest galactose and highest copper concentrations are labeled) and the corresponding quartic potential (right). (**D**) At the galactose concentration of 250 μM, the unstable equilibrium $f_{eq}$ is close to 1/2. Different technical replicates that start around $f = 1/2$ tend toward different stable equilibria of **Equation 1** (i.e., $f = 0$ and $f = 1$) in the long-term limit, highlighting the instability of the equilibrium point. Yellow points show frequencies of the K1 strain 14 hr after inoculation, green points show frequencies of the K1 strain 38 hr after inoculation.

The online version of this article includes the following source data and figure supplement(s) for figure 3:

**Source data 1.** This Excel spreadsheet contains all data used to plot **Figure 3**.

**Figure supplement 1.** This figure shows the same data as **Figure 3B**, but with the predictions of the models with **Equation 11** (magenta) and **Equation 12** (cyan) instead of **Equation 1**, modified to account for toxin production by both strains, and parametrized using competition assays between the killer strains and sensitive strains (**Figure 2**): K1 versus S1 and K2 versus S1.

**Figure supplement 2.** This figure shows the frequency of the K1 killer strain at the first (horizontal axis) and second (vertical axis) measurement time point for different technical replicates of a competition experiment with 250 μM galactose, which is close to the unstable equilibrium and shows replicates that tend toward different stable equilibria in the limit of large times (the same data are plotted in **Figure 3D**).

---

the initial condition when determining which replicates tend toward different stable equilibria in the long-term limit. Overall, the experimental results from well-mixed experiments (**Figure 3**) confirm the theoretical prediction that a critical starting frequency, the equilibrium frequency $f_{eq}$, is required for a stronger antagonist to invade a resident, antagonist population.

Because many microbial communities are spatially structured, we studied the spatial dynamics of antagonism in *spatially* structured populations growing on surfaces. As an intermediate step between well-mixed liquid cultures and spatially structured populations on surfaces, we studied the interaction between antagonistic strains on a solid surface; we distributed an initially well-mixed population of the two killer strains K1 and K2 on the surface of agar plates, with different concentrations of the inducers and with different initial frequencies of the two strains (**Figure 4**). We let the two strains grow for 24 hr and then measured their relative frequencies using a fluorescence stereomicroscope. These experiments were designed to investigate if the same inducer concentrations used in liquid led to similar strengths of antagonism between the two toxin-producing strains growing on the surface of agar plates. We found that increasing the concentration of the two inducers led to increased killing activity for the two strains, and that the copper-induced killer K2 appeared to be a more effective killer on plates than in liquid, as suggested by the fact that the equilibrium points $f_{eq}$ in liquid (**Figure 3B**) are smaller than those on plates at comparable concentrations of galactose (**Figure 4**), and by the observation that the competitive exclusion of K1 by K2 on plates with 12.5 μM copper happened much faster than in liquid with 50 μM copper (compare **Figure 4**, showing data after 24 hr from inoculation on plates, with **Figure 3B**, showing data after 38 hr from inoculation in liquid). Although we do not know why the K2 killer strain was a stronger antagonist on plates than in liquid, one possibility is that the agar (the only ingredient that differs between the liquid and solid media) contained traces of copper (**Debergh, 1983**) leading to a stronger expression of the K2 toxin genes.

## Invasion in spatially structured populations requires a critical inoculum size

Finally, we investigated antagonism dynamics in spatially structured populations. We asked whether an invading antagonist inoculated at one location could invade a surface uniformly occupied by a resident, weaker antagonist (see sketch in **Figure 1C**). We spread a uniform lawn of the weaker killer strain K2$_b$ on the surface of an agar plate and inoculated droplets of different volumes of a culture of strain K1. Experiments were performed with 360 μM galactose, an inducer concentration at which strain K1 is a stronger antagonist than strain K2$_b$. We let the two strains grow on the

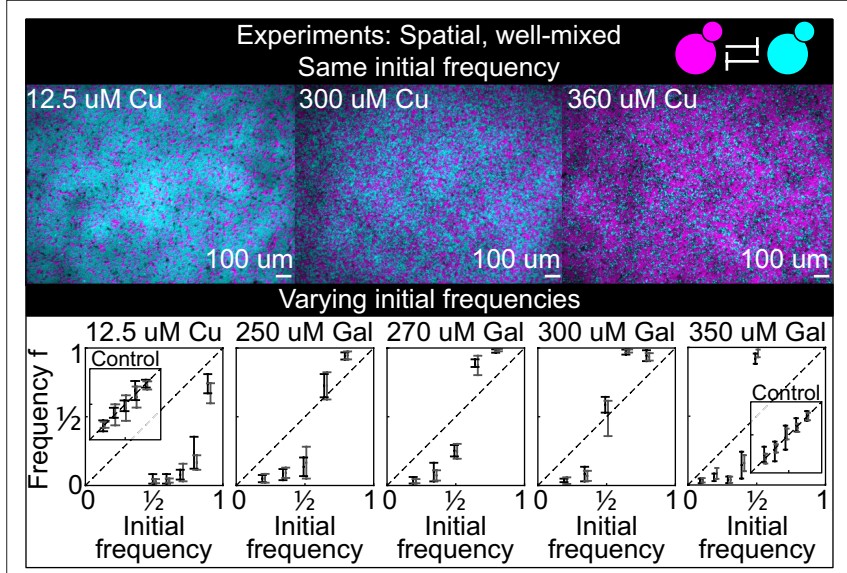

**Figure 4.** Antagonistic competition of spatially well-mixed, toxin-producing strains K1 and K2 growing on solid agar surfaces, at different inducer concentrations (subpanels) and different initial frequencies. The upper panels show combined fluorescence images of three representative spatially well-mixed populations imaged 5 hr after inoculation. The images show populations that started at 50:50 initial frequencies of K1 and K2, with K2 outcompeting K1 on the left and the opposite outcome on the right. The lower panels show the relative frequencies of the two strains at the time of inoculation (x axis) and 24 hr after inoculation (y axis), estimated as the relative fraction of space occupied by each strain. Black and gray data show data from two different experiments, and the two whiskers of each data point connect the maximum and the minimum estimated frequency $f$ of K1. Due to the difficulty of unequivocally assigning each pixel to one or the other strain in this assay, we report conservative estimates of the maximum and minimum frequencies that we can confidently assign to the K1 strain. The dashed lines show the 1:1 line that points would lie on if the two strains had equal fitness. Insets in the lower subpanels show control experiments: competition assays between the nonkiller strains S1 and S2, at the same inducer concentrations as the parent subpanels.

The online version of this article includes the following source data for figure 4:

**Source data 1.** This Excel spreadsheet contains all data used to plot **Figure 4**.

surface of these agar plates for 48 hr, following which a fraction of the populations was transferred onto fresh plates by replica plating, providing fresh nutrients while preserving their spatial structure (**Figure 5A**), and we allowed them to grow for further 48 hr. This procedure was repeated for 13 serial transfers. We imaged the populations at the end of each growth period (**Figure 5B**) using a fluorescence stereomicroscope, and measured the area occupied by the invading strain in each population (**Figure 5C–D**). As shown in **Figure 5B–D**, all K1 populations whose initial area was below 12 mm$^2$ were outcompeted by the K2$_b$ lawn and driven to extinction, whereas all those whose initial area was above 12 mm$^2$ managed to persist and eventually expanded displacing the resident K2$_b$ population (except for one outlier population highlighted in **Figure 5B–D** with a gray arrow and gray lines, which we discuss separately below). Most of the populations exhibited two characteristic phases in their spatial dynamics (**Figure 5C–E**): an initial retreat of both strains, leaving a region devoid of cells at the outer edge of the initial inoculum (black 'halos' in **Figure 5E**, of width 400±200 µm, mean ± SD), followed by the expansion of the K1 strain. All the inoculations below the critical inoculum size, instead, went extinct very rapidly (**Figure 5A and F**). A control experiment performed in parallel using the two sensitive, nonkiller strains S1 and S2 shows that all inoculations maintained their area following successive transfers (**Figure 5—figure supplement 2**). Thus, the retreat and expansion dynamics and the dependence of invasion success on the size of the initial inoculum can be attributed to the antagonistic interactions engineered between strains K1 and K2$_b$.

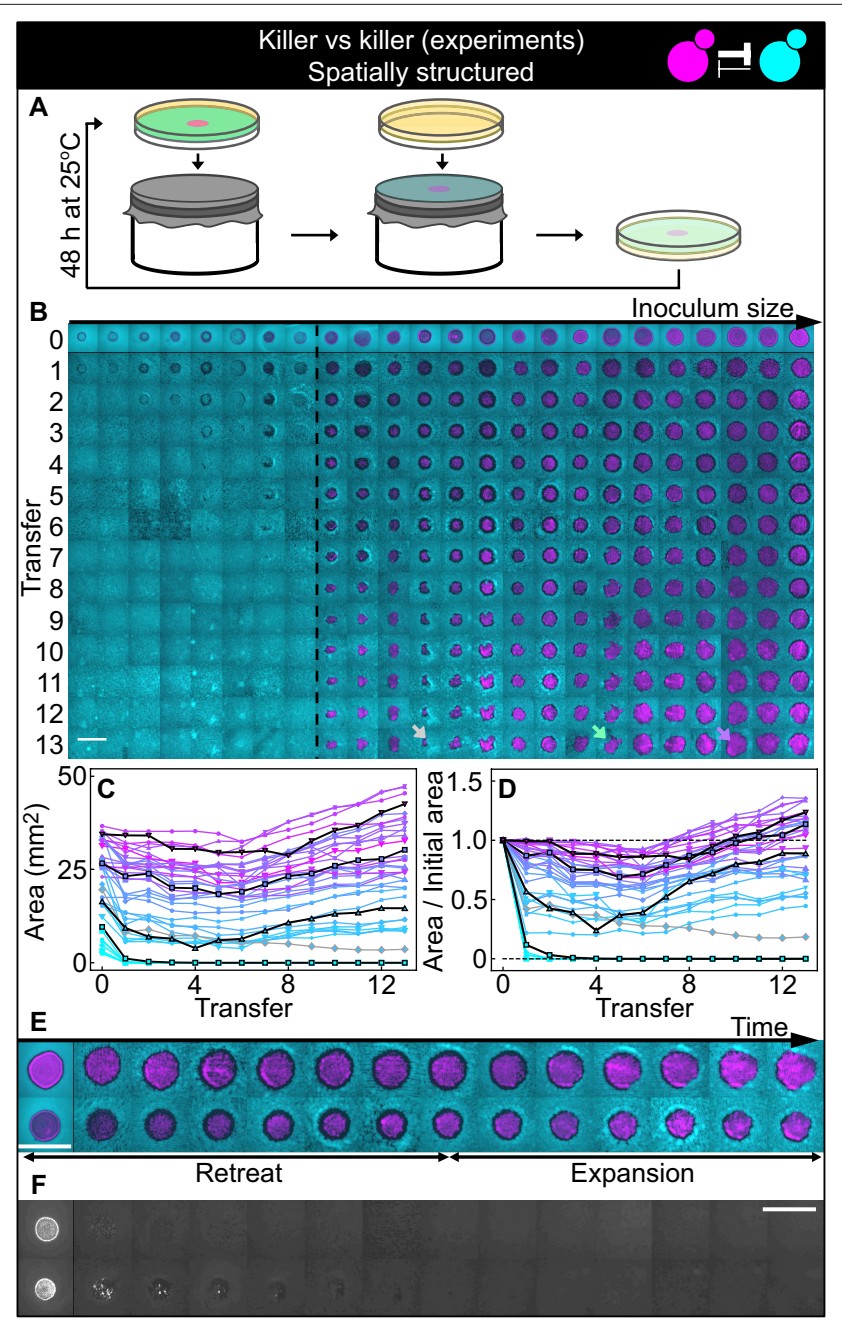

**Figure 5.** Antagonistic competition of killer strains K1 and K2$_b$ in spatially structured populations. Experiments were performed on agar medium with 360 μM galactose, a concentration at which the K1 strain is a stronger killer than the K2$_b$ strain. (**A**) We used replica plating to replenish nutrients and dilute the populations, while preserving the spatial structure of the population. At the end of each growth period (48 hr at 25°C), the agar plates hosting the experimental populations were gently pressed onto a microfiber cloth laid flat on a cylinder, leaving a diluted copy of the population on the cloth. A fresh agar plate was then pressed onto the same cloth, leading to an effective dilution of the population that preserved its spatial structure. (**B**) Shown from left to right are spatially structured populations of the invader K1 (magenta) and the resident K2$_b$ (cyan) strains originated from depositing droplets of different volumes of K1 onto a lawn of K2$_b$ cells. The populations are ordered based on the number of K1 cells at the end of the first growth cycle (estimated from the integrated ymCherry fluorescence of strain K1): the population with the smallest number of K1 cells is on the left, that with the largest number on the far right (first row). Different rows in the same column show the same population 48 hr after the previous transfer. The existence of a critical inoculum size is clearly visible and is marked by a dashed, black line. Populations on the left of this

*Figure 5 continued on next page*

*Figure 5 continued*

line failed to expand, whereas populations on the right of it persisted and eventually expanded. (**B**) shows only a subset of the experimental populations; ***Figure 5—figure supplement 1*** shows all the populations. An outlier population (gray arrow) and a population with a re-invading K2$_b$ sub-population (green arrow) are discussed in the main text. (**C**) Area covered by each K1 population at the end of each 48 hr growth period between transfers, color coded from cyan to magenta according to the integrated ymCherry fluorescence intensity of each replica at the end of the first growth period. Highlighted in black are four characteristic curves highlighting the fact that populations above the critical inoculum initially decrease in size, before expanding later. (**D**) Same data as in (**C**), divided by the initial area to highlight relative changes. Shown in (**E**) are two populations in which the retreat and expansion phases of the dynamics are clearly visible. The two rows show two different populations, whereas different columns show the same populations at the end of the growth periods following successive transfers. Both K1 (magenta) and K2$_b$ (cyan) populations initially retreat, leaving a region without cells (a black 'halo' surrounding the magenta islands). In the expansion phase of the dynamics, the magenta regions expand and increase their area. These populations are well above the critical inoculum size. Panel (**F**) shows the temporal dynamics of the two largest populations below the critical inoculum (last two columns of panel B before the dashed, black line). The largest population (first row in F) disappeared almost immediately, whereas the second-to-last one (second row) disappeared after five transfers. Only the fluorescence due to strain K1 is shown. Scale bars are 1 cm long. ***Figure 5—figure supplement 2*** shows a control experiment in which we followed the same protocol using the sensitive, nonkiller strains S1 and S2, and found that the areas of S1 inoculations on a lawn of S2 cells remained constant with time, for all initial inoculation sizes.

The online version of this article includes the following source data and figure supplement(s) for figure 5:

**Source data 1.** This Excel spreadsheet contains all data used to plot ***Figure 5***, ***Figure 5—figure supplement 1***, and ***Figure 5—figure supplement 2***.

**Figure supplement 1.** Shown here are all experimental replicates of the experiment of ***Figure 5***.

**Figure supplement 2.** Non-antagonistic competition of sensitive, nonkiller strains in spatially structured populations.

**Figure supplement 3.** A population of K1 cells (magenta) surrounded by K2$_b$ cells (cyan) imaged after 48 hr of growth at 25°C immediately before replica plating (**A**), immediately after replica plating (**B**), and after further 48 hr of incubation at 32°C (a temperature at which the K1 and K2 toxins are unstable) following replica plating (**C**).

**Figure supplement 4.** The toxins are unstable at high temperatures.

**Figure supplement 5.** The width of the halo grows with the toxin production rate by the two strains.

## Mutations in killer production and sensitivity alter the outcome of competitions

Our mathematical models ignores the possibility that mutations arise which alter the interaction between the antagonistic strains. Some replicates showed dynamics that differed from the typical dynamics described in the previous paragraph suggesting that such mutations occurred at detectable frequency. An outlier K1 population (K1°) highlighted with a gray arrow and gray lines in ***Figure 5B–D*** (first row of ***Figure 6A***) decreased its area monotonically with time, unlike all other populations that either went extinct or eventually expanded their area. At the end of this experiment, we collected cells from all the experimental populations and made glycerol stocks. Flow cytometry measurements of the fluorescent intensity of K1 cells sampled from the outlier replica K1° showed that their average fluorescent intensity was reduced with respect to the ancestral K1 population, and also compared to K1 cells sampled from a non-outlier population (K1$^s$) that successfully expanded in the spatially structured experiments (***Figure 6B***). Fluorescence imaging of colonies grown from the sampled and ancestor populations confirmed this observation. K1, K1$^s$, and K1° all contained both high- and low-fluorescence cells, but K1° contained many more low-fluorescence ones than K1 and K1$^s$. Because the gene expressing the fluorescent protein ymCherry is close to the K1 toxin gene on our genetic construct (***Figure 1A***), a hypothesis for the monotonic decrease in the area of the outlier population is that it was on average a weaker antagonist, expressing the K1 killer toxin gene to a reduced degree compared to the ancestor population and other populations that successfully expanded. Such a reduced expression may have occurred due to one of two sorts of mutation: a mutation that greatly elevated the rate of recombination between multiple, tandemly integrated copies of the construct, or a mutation that led to its reversible and epigenetic silencing.

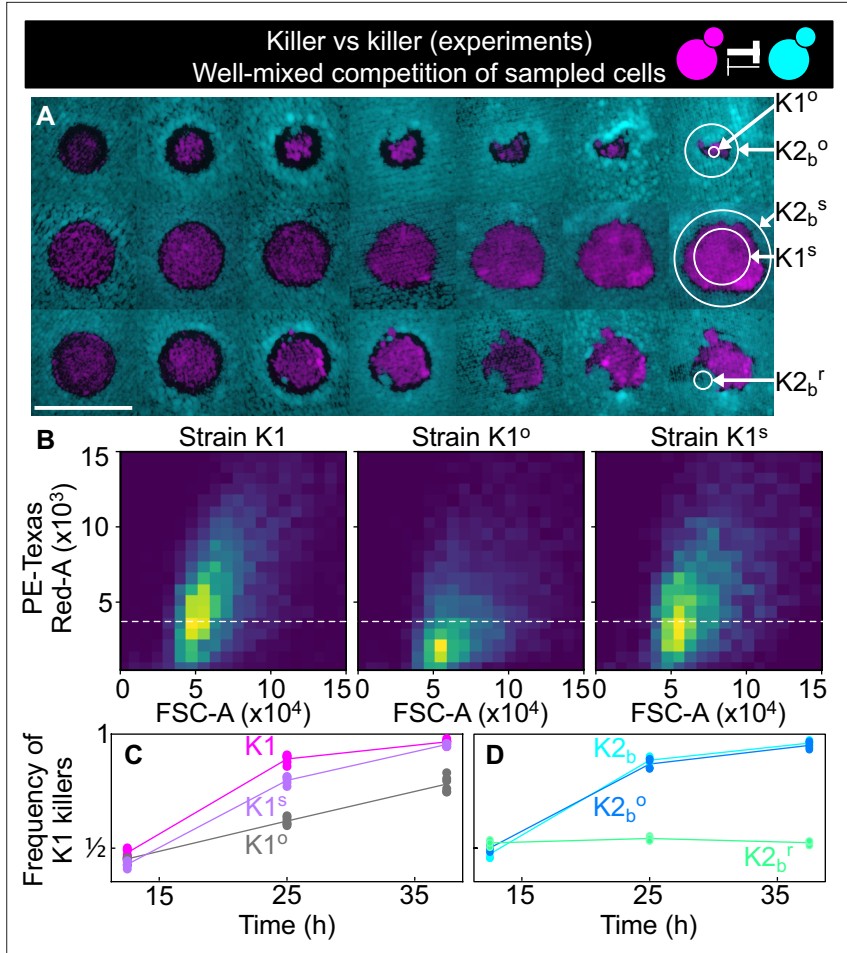

**Figure 6.** Cells sampled from certain replicates at the end of the experiment shown in *Figure 5* showed altered killing strength and toxin resistance. (**A**) Regions from which the samples were taken at the end of the experiments of *Figure 5*. The three rows show the replicates highlighted with gray, purple, and green arrows in *Figure 5B*, respectively. Different columns show the same populations at every other transfer. The scale bar is 1 cm long. (**B**) Density histograms of fluorescence intensity (y axis, arbitrary units) versus forward scatter (FSC, x axis, arbitrary units), which correlates with cell size, measured via flow cytometry during competitions in well-mixed liquid cultures between strains K1, K1°, and K1ˢ, versus K2ᵦ. K1 is the strain used in all other experiments, K1° and K1ˢ are populations sampled at end of the experiments of *Figure 5*. The fluorescence intensity of population K1° is lower than that of K1 and K1ˢ. The dashed line shows the K1 histogram mode as a visual aid to compare fluorescent intensities. (**C**) In competition assays in liquid at 360 μM galactose, the frequency of K1 and K1ˢ competing against strain K2ᵦ increases faster than the frequency of K1° competing against K2ᵦ, showing that population K1° is a weaker killer than K1 and of other populations that successfully expanded in the experiments of *Figure 5* (e.g., K1ˢ). (**D**) In competition assays in liquid at 360 μm galactose, strain K1 competing against strain K2ᵦ (cyan) and strain K1 against the sub-population K2ᵦ° (light blue) sampled at the end of the experiment of *Figure 5* follow similar dynamics suggesting that the collapse of K1° was not due to increased toxin production by K2ᵦ°, or it developing resistance to the K1 toxin. The competition assay with strain K1 against strain K2ᵦʳ (green), which re-invaded a K1 population in the experiments of *Figure 5*, instead, showed no increase in frequency for strain K1, suggesting that K2ᵦʳ developed resistance to the K1 toxin. Different data points in (**C–D**) show different technical replicates.

The online version of this article includes the following source data and figure supplement(s) for figure 6:

**Source data 1.** This Excel spreadsheet contains all data used to plot *Figure 6* and *Figure 6—figure supplement 1*.

**Figure supplement 1.** The population K1° isolated from the outlier replicate of *Figure 5B-D* is a weaker antagonist than the ancestor strain K1.

**Figure supplement 2.** Single-cell growth rates of the ancestor strains (K1, K2ᵦ) and of cells isolated at the end of the experiment of *Figures 5 and 6* (K1ˢ, K1°, K2ᵦ°, K2ᵦʳ) grown on YPD agar plates identical to the ones used for

*Figure 6 continued*

the experiments of *Figures 5 and 6*, but in the absence of inducers.

**Figure supplement 3.** Relative frequency of the ancestor strains (K1, K2$_b$) and of cells isolated at the end of the experiment of *Figures 5 and 6* (K1$^s$, K1$^o$, K2$_b^o$, K2$_b^r$) in competition against sensitive strains in well-mixed liquid cultures diluted daily in YPD medium, in the absence of inducers.

We tested the hypothesis that the outlier population K1$^o$ was a weaker antagonist by performing a competition assay in liquid medium, competing K1$^o$ against the ancestor K2$_b$ population and against the nonkiller strain S2, at 50:50 initial frequencies following the same protocol as the experiments of *Figures 2 and 3A*. In the same experiment, we competed the ancestor K1 and the successful invader, K1$^s$, against the ancestor K2$_b$ (*Figure 6C* and *Figure 6—figure supplement 1*, panel A) and against the nonkiller strain S2. We found that K1$^s$ and the ancestor K1 increased in frequency faster than K1$^o$, demonstrating that the outlier population K1$^o$ was indeed a weaker antagonist than K1$^s$ and K1, which followed similar trajectories. An alternative hypothesis for the outlier behavior of K1$^o$ would be that K2$_b$ cells in that region developed resistance to the K1 toxin. Even though visual inspection of the spatiotemporal dynamics of K2$_b$ in the outlier replica suggest that this might have happened (cyan cells rapidly re-invaded the magenta K1$^o$ population from the top, see *Figure 6A*), competition assays between K1 and K2$_b$ cells sampled from the area surrounding K1$^o$ (K2$_b^o$) rule out this hypothesis, given that they followed the same dynamics of competition assays between the original K2$_b$ stock and K1 (*Figure 6D* and *Figure 6—figure supplement 1B*). Given the small size of the K2$_b$ front that invaded the K1 population from the top (*Figure 6A*), however, it is possible that when sampling K2$_b^o$ cells from the region surrounding K1$^o$ we failed to isolate and test potentially toxin-resistant cells from the invading region. Other populations showed interesting phenotypes, like the one highlighted with a green arrow in *Figure 5B* (third row of *Figure 6A*). In this population, K2$_b$ cells (cyan) re-invaded the K1 population (magenta) after the halo had formed. By competing K2$_b$ cells from the sub-population highlighted with the green arrow (K2$_b^g$) in *Figure 5B* against the ancestor K1 in competition assays in liquid media with 360 µM galactose, we found that they had greatly decreased (possibly null) sensitivity to the toxin produced by K1 (*Figure 6D*), suggesting that they became K1 resistant during the course of the experiment. *Figure 6—figure supplements 2 and 3* reveal that we could not detect any differences in growth rate between any pairs of strains, ruling out the possibility that the altered outcomes of competition observed in *Figure 6* could be due to changes in cell division times during the experiment.

## Models suggest that nutrient depletion produces unoccupied halos between antagonistic strains

The frequency model used for well-mixed populations cannot be directly applied to describe the experiments with spatially structured populations on surfaces, since it does not include the diffusion of the two toxins, which likely underlies features such as the halo region at the boundary between two antagonistic strains (*Figure 5E*). To mathematically investigate the spatial dynamics of the two antagonistic strains, we explored a suite of models in which we explicitly modeled the density of each strain (rather than their relative frequency as in *Equation 1*) as a function of space and time, along with the concentration of the two toxins. We used the parameters for toxin production derived from competition between killers and nonkiller strains (data shown in *Figure 2*) and estimated the diffusion rates of the toxins and yeast cells based on their size (Table 6). By exploring models with different levels of complexity and realism (Materials and methods), we found that we had to explicitly model the dynamics of nutrients (glucose) to reproduce the formation of the halo, which in the models consists of a region of mutual destruction, with significantly reduced cell density at the boundary between the two strains. In the most realistic model we investigated (Materials and methods), cells occupy a two-dimensional surface at the top of the agar plate and their growth dynamics is modeled via a growth rate that depends on nutrient concentration and saturates with an effective half-saturation constant, $K_m$, for the local nutrient concentration, and a death term proportional to the local concentration of the toxins. Cells diffuse locally on the surface of the agar via a growth-dependent diffusion term reflecting the fact that cells push each other around as they interact mechanically with other cells during their growth and division (*Giometto et al., 2018*; *Kayser et al., 2018b*). The toxins and the nutrients diffuse in the agar and are produced and/or depleted by cells at the surface, at rates that depend on

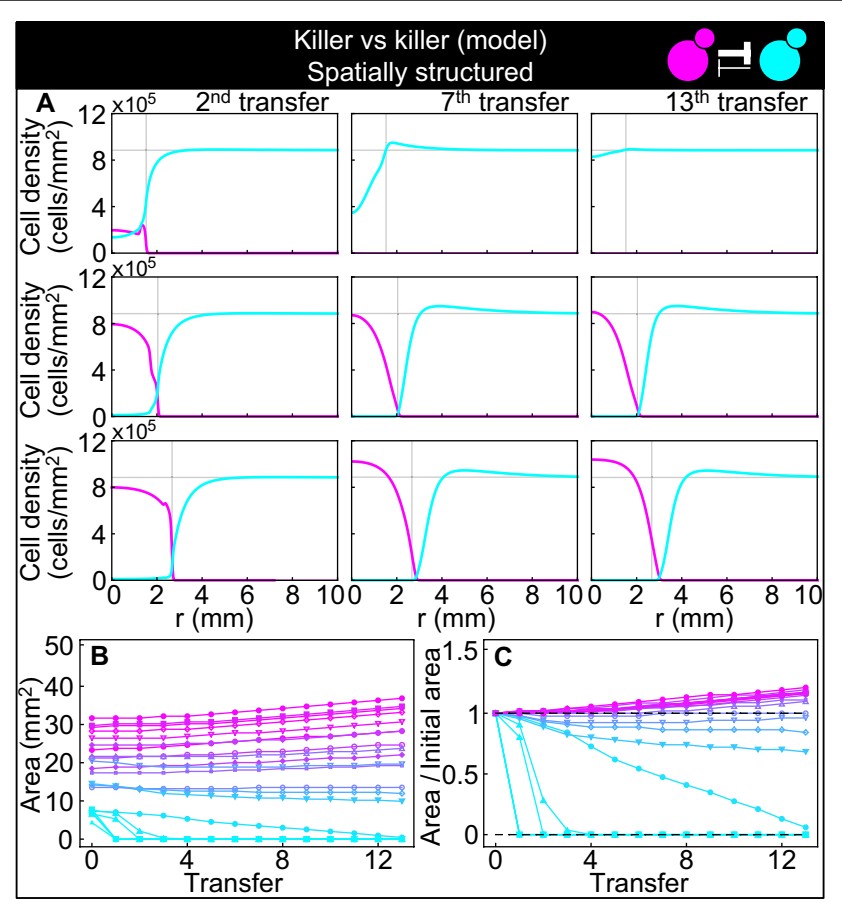

**Figure 7.** Numerical integrations of the spatial model (*Equations 7–9*), parametrized using the experiments of *Figure 2*, and other values taken from the literature, can qualitatively reproduce the experimental dynamics of antagonistic competition between strains K1 and K2$_b$ in spatially structured populations. (**A**) Simulated K1 inoculations smaller than the critical inoculum (first row) expanding on a landscape occupied by strain K2$_b$ fail to establish and expand, whereas larger inoculations do (second and third row). The model can reproduce the formation of the halo, a region without cells at the interface between the two strains (second and third row). (**B**) Area covered by each simulated K1 population expanding on a landscape occupied by strain K2$_b$ at the end of each 48 hr growth period between simulated transfers, color-coded from cyan to magenta according to the total K1 population of each replica at the end of the first growth period. (**C**) Same data as in (**B**), divided by the initial area to highlight relative changes. Compare (**B**) and (**C**) to *Figure 5C and D*.

The online version of this article includes the following figure supplement(s) for figure 7:

**Figure supplement 1.** Models that do not account for nutrient diffusion fail to reproduce the formation of the halo between two antagonist strains.

**Figure supplement 2.** Numerical integrations of the spatial model (*Equations 7–9*), parametrized using the experiments of *Figure 2* and other values taken from the literature, can qualitatively reproduce the experimental dynamics of antagonistic competition between strains K1 and K2$_b$ in spatially structured populations.

**Figure supplement 3.** Dependence of critical radius size on model parameters.

**Figure supplement 4.** The width of the halo and the shape of the two strains' density profiles varies with the parameters of the model.

---

the local density of the two strains. We found that this model, parametrized using the competition assays between toxin-producing and nonkiller strains and using suitable estimates for the diffusion coefficients of the nutrients, toxins, and yeast cells taken from the literature, could reproduce the striking halos observed at the boundary between the two strains (*Figure 7A*), suggesting that the halo emerges due to a combination of toxin-induced killing and diffusion of nutrients away from the agar beneath the halo and their consumption by the cells bordering the halo. Upon numerically integrating

the model, starting from initial conditions comparable to those used in the experiments with spatially structured populations, we found that it correctly predicts the extinction of the smaller inoculations, with a critical inoculum size between 8 and 14 mm$^2$, and the initial retreat and subsequent expansion of the larger ones (*Figure 7A–C*). The model also predicts an enhanced density of the two killer strains at the two sides of the halo (*Figure 7A*), caused by the diffusion of glucose away from the agar underneath the halo to sustain cell growth at the exposed edges of the regions occupied by the two strains. The enhanced density in those regions was seen also in the experiments as an increased fluorescence signal (see *Figure 5E*), even though the magnitude of this phenomenon appears larger in the experiments than in the model. If the halo was caused by the presence of the toxins alone, and not by the combined effect of the toxins and the diffusion of nutrients away from the agar underneath the halo, one would expect that inhibition of the toxin would allow cells to re-invade the halo region. To test this, we experimentally verified that no further growth in the halo region is observed after transferring populations that competed for 48 hr at 25–32°C for further 48 hr (*Figure 5—figure supplement 3*), a temperature at which both the K1 and K2 toxins are unstable and fail to inhibit the growth of susceptible strains (*Marquina et al., 2002*; *Lukša et al., 2016*, *Figure 5—figure supplement 4*).

## Discussion

We constructed an experimental system to study the dynamics of antagonism between yeast strains that produce and release two different toxins. We used this system to study the dynamics of antagonism in zero-dimensional well-mixed, two-dimensional well-mixed, and two-dimensional spatially structured populations. We derived mathematical models to describe each of these scenarios, parametrized the models using competition assays between killer and nonkiller strains, and showed that these models could predict the dynamics of antagonistic competition between toxin-producing strains. We verified the theoretical prediction that a critical inoculum size is required for a stronger antagonist to invade and displace a resident population of a weaker antagonist, in all of the spatial and non-spatial scenarios we tested.

The experiments in which we inoculated small populations of a stronger antagonist on a landscape occupied by a weaker one (*Figure 5*) revealed two unexpected features of antagonistic dynamics that had not been theoretically considered or predicted before. First, the expansion of the stronger antagonist was preceded by a contraction phase in which a halo devoid of cells formed between the two antagonists, both of which initially retreated. The existence of a halo had been observed before and used as a readout for killer activity by yeast geneticists (*Tipper and Bostian, 1984*), but its temporal dynamics had not been investigated from the point of view of population dynamics. Our investigation of increasingly realistic spatial models suggests that the diffusion of nutrients away from the boundary between two antagonistic strains plays an important role in the formation of the halo. Following its initial retreat, as the halo formed, the stronger antagonist started to expand, while the weaker antagonist kept retreating. Second, we found that occasionally new phenotypes emerged in the experiments after repeated transfers of the populations. Specifically, cells of the weaker, resident antagonist, resistant to the toxin produced by the stronger one, emerged in the population and formed fronts that re-invaded the population of the stronger antagonist. We believe that resistant cells were able to cross the nutrient-depleted region of the halo because, right after the populations were diluted by replica plating, resistant cells could grow and divide despite the presence of the invader's strain toxin, and they could thus take up nutrients located in that region of space before those nutrients diffused away. We also observed a population of the stronger antagonist with reduced killing activity, showing that the strength of the antagonistic interaction varied throughout the experiments and raising the interesting question of how such a mutant succeeded in outcompeting its ancestor which made more toxin. A recent study found that a strain infected with the K1 killer virus first lost the ability to produce the toxin, and later lost immunity to the K1 toxin during an evolutionary experiment which lasted for 1000 generations in well-mixed liquid cultures (*Buskirk et al., 2020*). In that context, immunity to the toxin was lost once the killer toxin was not present in the medium anymore. In our experiments, the outlier population K1° reduced its killing activity while the toxin produced by the resident strain was still present, suggesting that the evolutionary dynamics of microbial antagonism is quite rich and includes features that we do not fully understand.

Our results for yeast antagonism may also have implications for interactions between two antagonistic strains carried on the surface of liquid substrates. When confined at air-liquid interfaces due

to capillary forces, the metabolism of *S. cerevisiae* growing on a viscous liquid can produce density changes that generate fluid flows many times larger than their unperturbed colony expansion speed. That flow, in turn, can dramatically impact colony morphology and spatial population genetics (*Atis et al., 2019*). In these situations, an energetic cost associated with the interface between the area occupied by cells and the surrounding liquid (line tension) can play an important role, especially when a metabolically induced flow beneath the colony leads to the extrusion of thin strands of the colony (a fingering instability). For such surface-borne communities, the combination of fluid flow and line tension can have a profound effect on genetic outcomes. Our antagonistic yeast strains could be used to ask what happens to range expansions on liquid or solid substrates when the density dependence of killing generates a form of line tension for each strain and antagonism generates halos of destruction between them.

There is growing interest in designing synthetic microbial consortia made of different microbial strains that interact to produce various tasks (*Boynton, 2019*; *Kong et al., 2018*). From the engineering perspective, these synthetic consortia may be valuable tools to manipulate microbiomes for environmental and health-related applications. For example, using antagonistic interactions has been proposed as a strategy for antimicrobial intervention (*Gonzalez et al., 2018*). Synthetic consortia with predefined social interactions and some degree of control over the strength of such interactions (usually stepwise by using different promoters) have been designed using for example the prokaryotes *Lactococcus lactis* (*Kong et al., 2018*) and *Escherichia coli* (*Celik Ozgen et al., 2018*), and commensalism and cooperation have been designed in *S. cerevisiae* (*Müller et al., 2014*), where the strength of the interaction has been controlled in a continuous way by varying the availability of shared resources in the extracellular medium. Our work provides a synthetic model of microbial antagonism in which the interaction strength of two antagonist strains can be controlled independently and continuously, via titratable induction of promoters that respond to different chemicals. There are other killer yeast viruses and corresponding toxins available (*Liu et al., 2015*; *Magliani et al., 1997*; *Schmitt and Breinig, 2006*), as well as inducible promoters that respond to chemicals other than copper and galactose (*Lindstrom and Gottschling, 2009*; *Sangsoda et al., 1985*), leaving the possibility of expanding this system to more than two antagonist strains in the future. The ability to use engineered microbes to understand the fundamental features of antagonistic dynamics in simple spatial and non-spatial settings will help to model and design antagonistic interactions that could be exploited in synthetic microbial consortia, for example, to ensure that a strain can invade a pre-existing consortium and persist therein.

Our results may be of interest beyond the spatial dynamics of antagonistic microbial populations. Laboratory experiments with yeast and bacteria growing on surfaces have increased our understanding of ecological and evolutionary dynamics in dense microbial populations and biofilms (*Giometto et al., 2018*; *Hallatschek et al., 2007*; *Kayser et al., 2018a*; *Kayser et al., 2018b*), but also have implications for other dense cellular populations such as tumors growing in three dimensions (*Lamprecht et al., 2017*; *Lavrentovich and Nelson, 2015*). Cancers made of heterogeneous clonal populations, in particular, display a rich set of interactions among clonal sub-populations including one-way antagonistic interactions in which one clone inhibits others (*Marusyk and Polyak, 2010*), a situation analogous to the one considered in the experiments of *Figure 2*. Of course, the interaction dynamics and geometry of heterogeneous clonal populations in cancer is much more complex than the experiments performed here. The spatial dynamics of antagonistic populations (*Lavrentovich and Nelson, 2019*), as well as those of gene drives (*Tanaka et al., 2017*), and of hybrid zones (*Barton and Hewitt, 1985*; *Rouhani and Barton, 1987*; *Szymura and Barton, 1986*) all share analogies with classical nucleation physics. For all these spatial processes, one can define a potential energy function (*Figure 3C*) with two minima corresponding to stable points in which one variant competitively excludes the other one, separated by a finite energy barrier that defines a critical inoculum size required to establish a successful, expanding invasion that drives the interacting system from one state to the other. In the case of gene drives, such an energy barrier may help preventing the accidental spread of these genetic constructs (*Tanaka et al., 2017*), which have the potential of damaging natural ecosystems.

Finally, we note that our mathematical models predict the behavior of idealized and unchanging populations, whereas organisms acquire mutations and selection on these mutations can produce results that violate simple, theoretical predictions. In our experiments, we saw the appearance of mutations that altered the outcome of competitions by either changing the rate of toxin production

or the sensitivity to the toxin secreted by an antagonistic strain. In the natural world, the appearance and interactions of such mutants is likely to play a substantial role in the long-term behavior of antagonistic populations, especially in spatially structured populations where the descendants of the original mutant cell remain close to each other and can reap collective benefits from their altered behavior.

# Materials and methods

**Key resources table**

| Reagent type (species) or resource | Designation | Source or reference | Identifiers | Additional information |
|---|---|---|---|---|
| Strain, strain background (*Saccharomyces cerevisiae*) | W303 | Murray lab | W1588 | The complete list of derived strains is available in *Table 2* |
| Chemical compound, drug | Copper(II) sulfate pentahydrate | Sigma Aldrich | C8027 | |
| Chemical compound, drug | D-(+)-Galactose | VWR | VWRV0637 | |
| Chemical compound, drug | Ethylene glycol-bis(β-aminoethyl ether)-*N,N,N′,N′*-tetraacetic acid (EGTA) | Sigma Aldrich | E34378 | |
| Software, algorithm | Fiji (ImageJ) | PMID:22743772 | | |
| Software, algorithm | FlowCytometryTools | https://eyurtsev.github.io/FlowCytometryTools/, (*Friedman and Yurtsev, 2013*) | | Version 0.5.0 |
| Software, algorithm | Mathematica | Wolfram | SCR_014448 | Version 12.1.0.0. Custom scripts available at https://github.com/andreagiometto/Giometto_Nelson_Murray_2020 |
| Software, algorithm | Matlab | Mathworks | RRID:SCR_001622 | Version R2018a. Custom scripts available at https://github.com/andreagiometto/Giometto_Nelson_Murray_2020 |
| Software, algorithm | Python | Python | RRID:SCR_008394 | Custom scripts available at https://github.com/andreagiometto/Giometto_Nelson_Murray_2020 |

## Genetic constructs

The killer toxin and fluorescent protein genes used in this study were cloned into integrative plasmids. The K1 killer toxin gene was PCR-amplified from the plasmid YES2.1/V5-HIS-TOPO-K1 pptox (*Breinig et al., 2006*; *Gier et al., 2017*), which contains a DNA copy of the region of the dsRNA M1 virus genome encoding for the K1 toxin, and for the immunity of the host cell to the toxin, followed by the terminator $T_{CYC1}$. We obtained YES2.1/V5-HIS-TOPO-K1 pptox from Manfred Schmitt and Frank Breinig. The primers used for the amplification were oAG24 and oAG25, which were designed to clone the $P_{GAL1}$ promoter in front of the K1 killer toxin gene via Gibson assembly, and to keep $T_{CYC1}$ at the end of the K1 killer toxin gene. The K2 killer toxin gene was amplified via PCR from a pUC-based plasmid containing a DNA copy of the region of the dsRNA M2 virus genome that encodes for the K2 toxin (*Serviene et al., 2007*), and for the immunity of the host cell to the toxin. We obtained this pUC-based plasmid from Elena Servienė. The primers used were oAG26 and oAG29, which were chosen to clone the $P_{CUP1}$ promoter in front of the K2 killer toxin gene and the terminator $T_{CYC1}$ after it via Gibson assembly. The terminator $T_{CYC1}$ was amplified via PCR from YES2.1/V5-HIS-TOPO-K1 pptox using primers oAG25 and oAG31. The promoter $P_{GAL1}$ was amplified via PCR with primers oAG22 and oAG23 from a pFA6a-*kanMX6*-$P_{GAL1}$ plasmid (*Longtine et al., 1998*; *Wach et al., 1994*). The promoter $P_{CUP1}$ was amplified via PCR with primers oAG27 and oAG28 from a pFA6a-$P_{CUP1}$-*UBI-DHFR* plasmid. The segments containing $P_{GAL1}$ and K1-$T_{CYC1}$ were assembled via Gibson assembly (plasmid pAG11), using as backbone a pFA6a-*prACT1-ymCherry-KanMX6* plasmid (pAG3) linearized with the restriction enzymes *EcoRI* and *EcoRV*. The segments containing $P_{CUP1}$, *K2*, and $T_{CYC1}$ were assembled via

Gibson assembly (plasmid pAG14), using as backbone a pFA6a-*prACT1-ymCitrine-KanMX6* plasmid (pAG5) linearized with the restriction enzymes *EcoRI* and *EcoRV*. These plasmids were linearized at the $T_{CYC1}$ locus using the restriction enzyme *PpuMI*, and their integration at the $T_{CYC1}$ locus was verified using colony PCR using primers oAG5/oAG46 and oAG44/oAG45 (strain K1), and oAG8/oAG46 and oAG44/oAG45 (strains K2 and K2$_b$).

## Strains

The strains used here were derivatives of the *S. cerevisiae* strain yJHK234 derived from the W303 genetic background. This strain was constructed as described in *Ingolia, 2006*; *Ingolia and Murray, 2007* such that expression from $P_{GAL1}$ occurs in a titratable, unimodal way in response to changes in the extracellular concentration of galactose, because galactose has been turned into a gratuitous, non-metabolizable inducer by deleting the genes *GAL1* and *GAL10* from its genome, and the bistability in galactose induction has been removed by placing the *GAL3* gene under the constitutive promoter *PACT1* (*Ingolia, 2006*). By competing yJHK234 against reference K1 (strain F166), K2 (strain EX73), K28, and nonkiller K⁻ strains obtained from Manuel Ramírez (*Maqueda et al., 2012*), we found that yJHK234 has a K1⁺ phenotype, being resistant to the toxin produced by the reference K1 strain, sensitive to the toxins produced by the reference K2 and K28 strains, and capable of killing the reference nonkiller K⁻ strain. For our purposes, we needed an ancestor strain cured of the M1 virus, which could serve as an ancestor for both the killer, toxin-producing strains and for the sensitive, nonkiller ones. To obtain clones cured of the M1 virus, we spread about 100 cells on YPD agar plates with pH 4.5% and 0.001% (w/v) methylene blue (an indicator for cell death), incubated at 25°C for 24 hr and isolated blue colonies (clones cured of the virus being killed by surrounding non-cured colonies). We tested that the isolated clone yAG74 was sensitive to toxins secreted by both the K1 and K2 reference strains. To prevent catabolite repression, which would prevent expression of the K1 toxin from $P_{GAL1}$ in the presence of glucose, we deleted the hexokinase two gene, *HXK2* (*Raamsdonk et al., 2001*) from yAG74 (leading to yAG75) via transformation of the *HphMX4* marker from a pFA6-*HphMX4* plasmid with 5′ and 3′ flanking sequences of *HXK2*, using primers oAG13 and oAG14. We checked the deletion using colony PCR with primers oAG15/oAG17 and oAG15/oAG39. The K1 killer strain (yAG94) was obtained by transforming the linearized plasmid pAG11 into strain yAG75. The K2 (yAG83) and K2$_b$ (yAG82) killer strains were obtained by transforming the linearized plasmid pAG14 into strain yAG75. The transformant clones yAG94 and yAG83 were chosen for the experiments because of the bright fluorescence signal of the transformed cells observed at the flow cytometer and at the stereo-microscope compared to other transformants, which suggests multiple integrations of the plasmid into the genome. Conversely, yAG82 was selected because of the weaker fluorescent signal of the transformed cells compared to yAG83, suggesting a single integration of the plasmid. The sensitive, nonkiller strains S1 (yAG96) and S2 (yAG99) were obtained by transforming strain yAG75 with the linearized plasmids pAG3 and pAG5 digested with the restriction enzyme *AgeI* (which cleaves DNA in $P_{ACT1}$), respectively.

## Media and growth conditions

All experiments were performed using YPD buffered at pH 4.5 and supplemented with adenine, tryptophan, and ethylene glycol-bis(β-aminoethyl ether)-*N*,*N*,*N'*,*N'*-tetraacetic acid (EGTA), a chelating agent that we used to reduce the baseline expression of $P_{CUP1}$. The medium was prepared by mixing 990 mL of Millipore-purified water, 20 g of BD Bacto Peptone, 10 g of BD Bacto Yeast Extract, 10 mL of a 1% (w/v) solution of adenine and tryptophan, 1.04 g of NaOH and 9.51 g of EGTA. Then, 2 g of NaOH were added to bring the EGTA into solution. Then, we added 11.2 g of succinic acid and brought the pH to 4.5 by adding approximately further 2 g of NaOH. The solution was then filter-sterilized. Agarose medium was prepared following the same procedure but using 590 mL of water instead of 990 mL. Separately, 400 mL of Millipore water were mixed with 20 g of BD Bacto Agar and microwaved for 2 min. The two solutions were then combined and used to fill Petri dishes. Solutions of copper(II) sulfate and of galactose at different concentrations were added to the media in different volumes according to the desired final concentration of the two inducers *Table 1 Table 2*.

**Table 1.** Oligos used in this study.

| Oligo name | Oligo sequence |
| --- | --- |
| oAG5 | ATTACAGCGTGCCACAGATG |
| oAG13 | AATTCTCCACACATAATAAGTACGCTAATTAAATAAAATGCGTACGCTGCAGGTCGAC |
| oAG14 | ACCTTCTTGTTGTTCAAACTTAATTTACAAATTAAGTTTAATCGATGAATTCGAGCTCG |
| oAG15 | TTCGCCACTGTCTTATCTAC |
| oAG17 | CCCGTGAATTTCTAACAAAG |
| oAG22 | ATCCAGTTTAAACGAGCTCGGTAAAGAGCCCCATTATCTTAGC |
| oAG23 | ACTTGGGTTGGCTTCGTCATGTTTTTTCTCCTTGACGTTAAAG |
| oAG24 | TAACGTCAAGGAGAAAAAACATGACGAAGCCAACCCAAGT |
| oAG25 | AGGCCACTAGTGGATCTGATAGCTTGCAAATTAAAGCCTTC |
| oAG26 | AAAACAAACTGTAACGAATTATGAAAGAGACTACCACCAG |
| oAG27 | ATCCAGTTTAAACGAGCTCGCATTACCGACATTTGGGCGC |
| oAG28 | CTGGTGGTAGTCTCTTTCATAATTCGTTACAGTTTGTTTTTCTTAATATC |
| oAG29 | GAAGCTCGCCCCTTAGATCTGATTTATATCCTATCCTAGCCGC |
| oAG31 | GGCTAGGATAGGATATAAATCAGATCTAAGGGCGAGCTTC |
| oAG39 | CCAGATGCGAAGTTAAGTGC |
| oAG44 | CTTTGGTGGGTTGAAGAAGG |
| oAG45 | GATCAATCTCTTGCAGCCAC |
| oAG46 | AAGCGATGATGAGAGCGACG |

## Competition assays

Competition assays in liquid media were performed as follows. Strains were plated from the glycerol stock 4 days prior to the start of the experiment and grown for 2 days at 30°C in YPD plates. One day before the start of the competition assays, overnight cultures were started by transferring cells from the plates to a tube containing 2 mL YPD buffered at pH 4.5, which was placed in a rotating roller drum at 30°C. At the start of the competition assays, 200 μL from the overnight cultures were centrifuged, the supernatant was removed, and cells were then resuspended in 2 mL autoclaved water. The centrifugation and resuspension were repeated twice to dilute away any toxins produced overnight. For killer-versus-nonkiller competition assays, the strains K1, K2, and K2$_b$ were mixed with strains S2, S1, and S1, respectively (so that the two strains expressed different fluorescent proteins). Cell suspensions containing different strains were then mixed at the desired relative frequencies, and 40 μL of these were then diluted in 10 mL YPD buffered at pH 4.5 with EGTA. Each replica in the competition assays consisted of 500 μL of this solution placed in deep-welled (capacity 2 mL/well), 96-well, round-bottomed plates, taped to a roller drum rotating at a frequency of 1 rotation per second and placed in a room kept at 25°C. Technical replicates were assigned to random positions on the 96-well plate, irrespective of the treatment they belonged to. At regular time intervals, small samples (≤10 μL) were taken from each well, diluted in 50 mM Tris-HCl, pH 7.8, and the relative frequencies of the two strains were measured by flow cytometry. Flow cytometry data was performed using the Python package FlowCytometryTools and custom Python and Mathematica scripts. Occasionally, during measurement with the flow cytometer, some wells were not measured due to the aspiration of bubbles by the robotic liquid handler that automatically measured the 96-well plates. Due to the temporal sensitivity of the assay, relative frequency data from those replicates could not be recovered, and thus we excluded those technical replicates from the analysis. Competition assays with the unusual cells sampled from the experiments of *Figure 5* (assays of *Figure 6*) were performed using the same

**Table 2.** Strains used in this study.

All strains are in the *Saccharomyces cerevisiae* W303 background. Strain yAG75 has not been used in the experiments but is reported here because it is the ancestor of all the other strains; only those elements that differ from yAG75 are listed for the other yAG strains.

| Strain | Genotype | Note |
|---|---|---|
| yAG75 | MATa BUD4-S288C can1-100 gal1/10Δ::LEU2 his3-11,15 $P_{GAL3}$Δ::HIS3-MX6-$P_{ACT1}$-GAL3 ura3Δ0 hxk2Δ::HphMX4 | Sensitive to killer toxins K1 and K2, ancestor of all other strains |
| yAG82 | yAG75, $P_{ACT1}$-ymCitrine-$T_{ADH1}$ $P_{TEF}$-KanMX6-$T_{TEF}$ $P_{CUP1}$-K2-$T_{CYC1}$ | Strain K2$_b$, yAG75+ pAG14 |
| yAG83 | yAG75, $P_{ACT1}$-ymCitrine-$T_{ADH1}$ $P_{TEF}$-KanMX6-$T_{TEF}$ $P_{CUP1}$-K2-$T_{CYC1}$ | Strain K2, yAG75+ pAG14 |
| yAG94 | yAG75, $P_{ACT1}$-ymCherry-$T_{ADH1}$ $P_{TEF}$-KanMX6-$T_{TEF}$ $P_{GAL1}$-K1-$T_{CYC1}$ | Strain K1, yAG75+ pAG11 |
| yAG96 | yAG75, $P_{ACT1}$-ymCherry-$T_{ADH1}$ $P_{TEF}$-KanMX6-$T_{TEF}$ | Strain S1, yAG75+ pAG3 |
| yAG99 | yAG75, $P_{ACT1}$-ymCitrine-$T_{ADH1}$ $P_{TEF}$-KanMX6-$T_{TEF}$ | Strain S2, yAG75+ pAG5 |
| F166 | MATα leu1 kar1 L-A-HNB M1 [K1$^+$] | Reference K1 strain |
| EX73 | MATa/α HO/HO L-A M2 [K2$^+$] | Reference K2 strain |

protocol as the competition described above, starting from glycerol stocks of the cells sampled from the experimental populations of **Figure 5**.

## Spatial experiments

The experiments with spatially well-mixed populations on surfaces shown in **Figure 4** were performed as follows; 28 mL of molten YPD agar with pH 4.5 and EGTA (*Media and growth conditions*) were added to 100 mm diameter Petri dishes 2 days before the start of the experiment, along with appropriate amounts of a 5 mM solution of copper(II) sulfate or a 50 mM solution of galactose to reach the desired target concentration of inducer on the plates. The day before the experiment we inoculated overnight cultures of strains K1, K2, S1, and S2 in 2 mL YPD culture tubes with pH 4.5 and 25 mM EGTA and grew them at 30°C on a rotating roller drum. At the start of the experiment, we centrifuged and resuspended 200 µL of the overnight cultures in 2 mL autoclaved Millipore-purified water, repeating the centrifugation and resuspension twice to remove toxins from the overnight cultures. We mixed strains K1 and K2 with K1 frequencies $f = 1/2$ , $3/5$ , $7/10$ , $4/5$, and $9/10$ for the treatment with 12.5 µM copper, with K1 frequencies $f = 1/5$ , $7/20$ , $1/2$ , $13/20$, and $4/5$ for the treatments with 250, 270, and 300 µM galactose, and with K1 frequencies $f = 1/10$ , $1/5$, $3/10$, $2/5$ and $1/2$ for the treatment with 350 µM galactose. For the control experiments (insets in **Figure 3B**), we mixed strains S1 and S2 with S1 frequencies $f = 1/10$ , $3/10$, $1/2$, $7/10$, and $9/10$ for both treatments (12.5 µM copper and 350 µM galactose). Five µL droplets of the mixed solutions were inoculated at different, random locations on a regular lattice on the surface of the agar. Plates were placed inside a plastic box with an open water Schott flask to provide humidity, in a room set at 25°C. When depositing a droplet from an overnight culture on a plate, a large fraction of the cells in the droplet end up at distributed at the outer boundary of the droplet due to the coffee-stain effect (**Deegan et al., 1997**). In this experiment, we imaged the interior of the droplets deposited on the agar surface, far from the coffee-stain ring. The experiments with spatially structured populations on surfaces shown in **Figure 5** and its figure supplements were performed similarly but imaging the entire droplets. At the start of those experiments, we centrifuged and resuspended 200 µL of the overnight cultures K1, K2$_b$, S1, and S2 in 200 µL autoclaved Millipore water, repeating the centrifugation and resuspension twice to remove toxins from the overnight cultures. For the experiments of **Figure 5**, 200 µL of the resuspended overnight K2$_b$ culture were spread on the surface of the agar using an inoculating loop. Then, using a micropipette, we deposited droplets of the resuspended overnight K1 culture on top of the K2$_b$ lawn, at random locations on a regular lattice, well separated from each other. We deposited droplets of six different volumes (0.5–3 µL with 0.5 µL increments), with seven replicates per volume, across multiple plates. Similarly for the experiments shown in **Figure 5—figure supplement 2**, 200 µL of the resuspended overnight S2 culture were spread on the surface of the agar using an inoculating loop, and droplets of volumes 0.5 , 2 , and 3 µL of the S1 culture were deposited on top of the S2 lawn, with seven replicates per volume, at random locations on a regular lattice. We imaged each population at the end of the each 48 hr growth period, immediately before its transfer to the next plate, using a fluorescence stereomicroscope, always at the same magnification and with the same exposure time. The areas of the invading populations shown in **Figure 5C–D** and **Figure 5—figure supplement 2B-C** were measured via image analysis using custom Fiji scripts and Mathematica notebooks.

## The frequency model

The starting point for the derivation of the frequency model **Equation 1** is the following set of equations for the densities of two toxin-producing strains ($n_1$ and $n_2$) and the concentrations of the two toxins they produce ($c_1$ and $c_2$ , respectively):

$$\begin{cases} \dfrac{dn_1}{dt} = gn_1\left(1 - \dfrac{n_1 + n_2}{K}\right) - d_1 n_1 c_2 \\[2mm] \dfrac{dn_2}{dt} = gn_2\left(1 - \dfrac{n_1 + n_2}{K}\right) - d_2 n_2 c_1 \\[2mm] \dfrac{dc_1}{dt} = a_1 n_1 - b\left(n_1 + n_2\right) c_1 \\[2mm] \dfrac{dc_2}{dt} = a_2 n_2 - b\left(n_1 + n_2\right) c_2 \end{cases} \tag{2}$$

where $n_i$ is the cell density of strain $i$, $g$ is the growth rate of the two strains in isolation (assumed to be identical for the two strains as they differ solely for the integrated genetic construct), $c_i$ is the concentration of toxin $i$, $d_i$ s are death rates per concentration of $c_j$ ($j \neq i$), and $a_i$ s and $b_i$ s are toxin production and toxin attachment rates (the two toxins bind to the same receptor on the cell wall of both sensitive and producer strains, and thus the term $n_1 + n_2$). *Equation 2* assumes that the growth of each strain can be described by a logistic growth term in which the carrying capacity is shared by the two strains. Toxin-induced cell death is introduced via a term proportional to the product between a strain's density and the concentration of the toxin produced by the other strain, and toxin production is assumed to be proportional to the density of the strain that produces it. Upon assuming that $c_1$ and $c_2$ are 'fast variables' (in a sense specified below) compared to $n_1$ and $n_2$, we set their temporal derivatives to zero to find the quasistatic approximations $c_1 = a_1 n_1 / [b(n_1 + n_2)]$ and $c_2 = a_2 n_2 / [b(n_1 + n_2)]$. After substituting these approximations in the first two lines of *Equation 2* and computing the temporal derivative for the fraction of strain one in the population, $f = n_1/(n_1 + n_2)$, we find:

$$\frac{df}{dt} = f(1-f)\left[\frac{a_1 d_2}{b}f - \frac{a_2 d_1}{b}(1-f)\right], \tag{3}$$

which is in fact *Equation 1* with the interaction coefficients $r_1 = a_1 d_2/b$ and $r_2 = a_2 d_1/b$. We can rewrite *Equation 3* as:

$$\frac{df}{dt} = (r_1 + r_2) f (1-f)(f - f^*), \tag{4}$$

with $f^* = r_2/(r_1 + r_2)$. *Equation 4* reveals that the dynamics of $f$ evolves with the characteristic time scale $\tau_f = 1/(r_1 + r_2)$. Once we indicate with $c_1^*$ and $c_2^*$ the quasi-fixed points $c_1^* = (a_1/b)f$ and $c_2^* = (a_2/b)(1-f)$, and letting $c_1(t) = c_1^* + \delta c_1(t)$ and $c_2(t) = c_2^* + \delta c_2(t)$, we can rewrite the last two lines of *Equation 2* as:

$$\begin{cases} d(\delta c_1)/dt = -b(n_1 + n_2)\delta c_1 \\ d(\delta c_2)/dt = -b(n_1 + n_2)\delta c_2 \end{cases} \tag{5}$$

which show that the toxin concentrations evolve with the characteristic time scale $\tau_{tox} = 1/(bn)$, with $n = n_1 + n_2$. Thus, for the quasistatic approximation to hold, we require that $\tau_{tox} \ll \tau_f$, that is, that $r_1 + r_2 \ll bn$. A best fit of *Equation 2* to the data on the competition between toxin-producing and sensitive, nonkiller strains allow us to check if this condition is met. For example, for the competition of K1 versus K2 with 360 μM galactose and 0 μM copper we have $r_1 = 0.13$ hr$^{-1}$, $r_2 = 0.04$ hr$^{-1}$ and $b = 1.2 \cdot 10^{-8}$ hr$^{-1}$(cell/mL)$^{-1}$ (see next section for its estimate), and thus the condition is satisfied for $n \gg (r_1 + r_2)/b \approx 10^7$ cells/mL, which we can compare to the carrying capacity $K = 2.3 \cdot 10^8$ cells/mL. With a starting density of about $3 \cdot 10^4$ cells/mL and a growth rate $g = 0.35$ hr$^{-1}$, it takes about 17 hr for the condition $n \gg (r_1 + r_2)/b$ to be met, and thus the frequency model is only strictly appropriate for describing the last few hours of the dynamics of our competition assays. This might explain why the frequency model tends to slightly overestimate the increase in frequency at $t = 14$ hr of the strongest antagonist strain in our competition assays, with respect to the data (*Figures 2A and 3A*). Nonetheless, the frequency model seems to do a good job in predicting the antagonistic dynamics between toxin-producing strains, possibly because the early phases of the dynamics are dominated by the exponential growth of the two strains and the toxins do not affect the dynamics too much at this stage. Strictly speaking, when the condition $n \gg (r_1 + r_2)/b$ is not met, the full model in *Equation 2* would be better suited to describe the data.

## Spatial models

Our starting point for the investigation of antagonistic population dynamics in spatially structured populations was the following spatial generalization of *Equation 2*:

$$\begin{cases} \dfrac{dn_1}{dt} = gn_1 \left(1 - \dfrac{n_1 + n_2}{K_{spatial}}\right) - d_1 n_1 c_2 + D_y \nabla^2 n_1 \\[2mm] \dfrac{dn_2}{dt} = gn_2 \left(1 - \dfrac{n_1 + n_2}{K_{spatial}}\right) - d_2 n_2 c_1 + D_y \nabla^2 n_2 \\[2mm] \dfrac{dc_1}{dt} = a_1 n_1 - b_1 \left(n_1 + n_2\right) c_1 + D_t \nabla^2 c_1 \\[2mm] \dfrac{dc_2}{dt} = a_2 n_2 - b_2 \left(n_1 + n_2\right) c_2 + D_t \nabla^2 c_2 \end{cases} , \tag{6}$$

where the constants $D_y$ and $D_t$ are the diffusion rates of the yeast cells and their toxins (the two toxins K1 and K2 have similar sizes, so we assumed that they have identical diffusion rates), and $K_{spatial}$ is the spatial carrying capacity (with units of cells/cm²), which is related to the well-mixed carrying capacity via the relationship $K_{spatial} = Kh$, where $h$ is the height of the medium in the agar plate. We used a Markov Chain-Monte-Carlo (MCMC) algorithm (**Vrugt et al., 2009**) to fit the non-spatial version of this model **Equation 2** to the data from the competition assays between toxin-producing and nonkiller strains. To reduce the number of free parameters, we assumed $b_1 = b_2$ and we partially nondimensionalized the equations through appropriate rescaling of the variables and parameters (section *Parameter fitting*). We found that the non-spatial version of this model could indeed fit the antagonistic dynamics between toxin-producing and nonkiller strains, and that it could predict the dynamics of antagonistic competition between the two toxin-producing strains K1 and K2 (and K1 versus K2_b) in well-mixed media (**Figure 3—figure supplement 1**). However, when we numerically integrated the model in an attempt to reproduce the dynamics that we had observed in the spatially structured experiments, **Equation 6** failed to reproduce the halo between the two toxin-producing strains (**Figure 7—figure supplement 1A**), at least within a reasonable range for the various parameters of the model. The failure of such model to reproduce the halo can be explained as follows. The logistic growth term in **Equation 6** assumes that every cm² on the agar can support $K_{spatial}$ cells. In such a model, nutrients located in a given region of space cannot diffuse to nearby regions and thus can only support the growth of cells locally. With toxin production rates representative of our experiments, the toxin produced by an antagonist strain is not sufficient to completely halt the growth of the other antagonist, as shown by the fact that the absolute number of cells of both antagonists grew in all our well-mixed competition experiments, even if the relative frequency of one of the strains declined with time. In the model with logistic growth, the two populations are thus able to grow at the interface between the two antagonist strains, even if at a slower pace compared to other regions of space, eventually almost completely filling the halo region with cells (**Figure 7—figure supplement 1A**). When nutrients can diffuse, however, nutrients move to other regions of space before cells at the interface between the two antagonists are able to grow, leading to the depletion region that we referred to as the 'halo'. Variants of **Equation 6** that accounted for a more realistic cell diffusion term (see discussion below **Equation 7**, for the fact that the toxin production rate is likely dependent on growth rate and for the fact that the toxins can diffuse into the agar below the surface on which cells are located also failed to reproduce the halo using biologically realistic parameters (**Figure 7—figure supplement 1B-D**)). These results motivated us to gradually increase the complexity and realism of the model to identify the processes that are responsible for the dynamics observed in the experiments. In the most realistic model we investigated, the equations for the two strains, inhabiting the two-dimensional surface at the top of the agar (at the height coordinate $z = h$), read:

$$\begin{cases} \dfrac{dn_1}{dt} = g_{max} n_1 \dfrac{c_g|_{z=h}}{c_g|_{z=h} + K_S} - d_1 n_1 c_2|_{z=h} + D_y \nabla \cdot \nabla \left(n_1 \dfrac{c_g|_{z=h}}{c_g|_{z=h} + K_S}\right) \\[3mm] \dfrac{dn_2}{dt} = g_{max} n_2 \dfrac{c_g|_{z=h}}{c_g|_{z=h} + K_S} - d_2 n_2 c_1|_{z=h} + D_y \nabla \cdot \nabla \left(n_2 \dfrac{c_g|_{z=h}}{c_g|_{z=h} + K_S}\right) \end{cases} \tag{7}$$

where $c_g|_{z=h}$ is the glucose concentration ($c_g$) at the agar surface ($|_{z=h}$), $K_S$ is Monod's half-saturation constant for the growth rate of *S. cerevisiae* on glucose, $g_{max}$ is the maximum growth rate of the two strains (attained in the limit of infinite glucose concentration), and $c_i|_{z=h}$ is the concentration of toxin at the agar surface. Compared to **Equation 6**, the diffusion term has been modified to reflect the fact that in our experiments the predominant contribution to the local diffusion of cells is not due to Brownian motion, but rather to the growth dynamics of mother cells giving rise to daughter cells in

their immediate surroundings, and to the excluded volume forces that cells exert on each other while growing and dividing (*Giometto et al., 2018*; *Kayser et al., 2018b*). To model this phenomenon, we took the local flux of cells to be proportional to the local growth rate (having a standard diffusion term in *Equation 7* does not alter the results significantly, but it seems to us more realistic to have a growth-rate-dependent diffusion term in the model). The dynamics of $c_1$, $c_2$, and $c_g$ are now governed by the diffusion equation within the agar ($0 \leq z < h$):

$$\partial c_1/\partial t = D_t \nabla^2 c_1, \ \partial c_2/\partial t = D_t \nabla^2 c_2, \ \partial c_g/\partial t = D_g \nabla^2 c_g, \tag{8}$$

where $D_g$ is the diffusion coefficient of glucose, complemented by the following fluxes at the agar surface $z = h$:

$$
\begin{cases}
D_t \left. \dfrac{\partial c_1}{\partial z} \right|_{z=h} = a_1 n_1 \dfrac{c_g|_{z=h}}{c_g|_{z=h} + K_S} - b\,(n_1 + n_2)\,c_1|_{z=h} \\[2ex]
D_t \left. \dfrac{\partial c_2}{\partial z} \right|_{z=h} = a_2 n_2 \dfrac{c_g|_{z=h}}{c_g|_{z=h} + K_S} - b\,(n_1 + n_2)\,c_2|_{z=h} \\[2ex]
D_g \left. \dfrac{\partial c_g}{\partial z} \right|_{z=h} = -\dfrac{g_{max}}{Y} \dfrac{c_g|_{z=h}}{c_g|_{z=h} + K_S}\,(n_1 + n_2)
\end{cases}
\tag{9}
$$

Here, $Y$ is the cellular yield (cells/g glucose), and we impose no-flux boundary conditions on all other surfaces (i.e., where the agar is in contact with the Petri dish plastic). We parametrized the non-spatial version of this model using an MCMC algorithm (*Vrugt et al., 2009*) and the data from the competition assays between the killer strains and the nonkiller strains S1 and S2. We found that the non-spatial version of this model could fit the antagonistic dynamics between toxin-producing and nonkiller strains, and that it could also predict the dynamics of antagonistic competition between the two toxin-producing strains K1 and K2 (and K1 versus K2$_b$) in well-mixed media.

## Parameter fitting

The parameters of the frequency model were obtained by least-squares fitting of the equation:

$$\frac{df}{dt} = rf^2\,(1 - f), \tag{10}$$

where $f$ is the frequency of the toxin-producer strain, to the data on the competition between toxin-producing and sensitive, nonkiller strains shown in *Figure 2*. Note that *Equation 10* is a special case of *Equation 1* and describes a toxin-producer strain competing against a nonkiller one. The best-fit parameters obtained via least-squares fitting are reported in *Table 3*.

The parameters of *Equation 2* were fit to the cell density data from the competition assays between toxin-producing and sensitive, nonkiller strains using MCMC (*Vrugt et al., 2009*). For a toxin-producing (K) strain competing versus a sensitive, nonkiller strain (S), *Equation 2* reads:

**Table 3.** Best-fit estimates for the parameters of the frequency model *Equation 10* fitted to the data from competition assays between toxin-producing strains and sensitive, nonkiller ones.
Concentrations of the inducers galactose (Gal) and copper (Cu) are indicated as superscripts, whereas the parameters $r_1$, $r_2$, and $r_{2b}$ (i.e., the parameter $r$ in *Equation 10* for the strains K1, K2, and K2$_b$) are given in units of 1 /hr (mean ± SD).

| $r_1^{0\mu M\ Gal}$ | $r_1^{200\mu M\ Gal}$ | $r_1^{240\mu M\ Gal}$ | $r_1^{280\mu M\ Gal}$ | $r_1^{320\mu M\ Gal}$ | $r_1^{360\mu M\ Gal}$ | |
|---|---|---|---|---|---|---|
| 0.005±0.002 | 0.013±0.008 | 0.020±0.002 | 0.038±007 | 0.090±0.004 | 0.125±0.014 | |

| $r_2^{0\mu M\ Cu}$ | $r_2^{12.5\mu M\ Cu}$ | $r_2^{25\mu M\ Cu}$ | $r_2^{50\mu M\ Cu}$ | $r_{2b}^{0\mu M\ Cu}$ | $r_{2b}^{5\mu M\ Cu}$ | $r_{2b}^{50\mu M\ Cu}$ |
|---|---|---|---|---|---|---|
| 0.037±0.007 | 0.078±0.004 | 0.092±0.002 | 0.109±0.002 | 0.021±0.005 | 0.039±0.003 | 0.059±0.002 |

**Table 4.** Best-fit estimates for the parameters of the full model with logistic growth *Equation 11* fitted to the data from competition assays between toxin-producing strains and sensitive, nonkiller ones, via Markov Chain-Monte-Carlo (MCMC).

Concentrations of the inducers galactose (Gal) and copper (Cu) are indicated as superscripts. The parameters $a_1$ and $a_2$ correspond to the rescaled parameter $a' = da$ of *Equation 11* for the strains K1 and K2, respectively.

| | | | |
|---|---|---|---|
| $g$ | $hr^{-1}$ | $a_1^{360\mu M\ Gal}$ | $2.36 \cdot 10^{-9}\,mL/cell/hr^2$ |
| $K$ | $2.3 \cdot 10^8\,cells/mL$ | $a_2^{0\mu M\ Cu}$ | $6.64 \cdot 10^{-10}\,mL/cell/hr^2$ |
| $b$ | $1.2 \cdot 10^{-8}\,mL/cell/hr$ | $a_2^{12.5\mu M\ Cu}$ | $1.43 \cdot 10^{-9}\,mL/cell/hr^2$ |
| $a_1^{0\mu M\ Gal}$ | $1.57 \cdot 10^{-10}\,mL/cell/hr^2$ | $a_2^{25\mu M\ Cu}$ | $1.64 \cdot 10^{-9}\,mL/cell/hr^2$ |
| $a_1^{200\mu M\ Gal}$ | $1.92 \cdot 10^{-10}\,mL/cell/hr^2$ | $a_2^{50\mu M\ Cu}$ | $1.92 \cdot 10^{-9}\,mL/cell/hr^2$ |
| $a_1^{240\mu M\ Gal}$ | $3.52 \cdot 10^{-10}\,mL/cell/hr^2$ | $a_{2b}^{0\mu M\ Cu}$ | $3.73 \cdot 10^{-10}\,mL/cell/hr^2$ |
| $a_1^{280\mu M\ Gal}$ | $7.32 \cdot 10^{-10}\,mL/cell/hr^2$ | $a_{2b}^{5\mu M\ Cu}$ | $6.55 \cdot 10^{-10}\,mL/cell/hr^2$ |
| $a_1^{320\mu M\ Gal}$ | $1.65 \cdot 10^{-9}\,mL/cell/hr^2$ | $a_{2b}^{50\mu M\ Cu}$ | $9.60 \cdot 10^{-10}\,mL/cell/hr^2$ |

$$\begin{cases} \dfrac{dn_K}{dt} = gn_K \left(1 - \dfrac{n_K + n_S}{K}\right) \\[2mm] \dfrac{dn_S}{dt} = gn_S \left(1 - \dfrac{n_K + n_S}{K}\right) - dn_S c \\[2mm] \dfrac{dc}{dt} = an_K - b\left(n_K + n_S\right)c \end{cases} \tag{11}$$

where $c$ is the toxin concentration. Not all parameters in *Equation 11* are identifiable, thus we rescaled the toxin concentration as $c' = dc$ and the toxin production rate as $a' = da$, which is formally equivalent to setting $d = 1$ in *Equation 11*, although the measurement units are affected as the rescaled toxin concentration has dimensions of 1/time. Apostrophes are dropped in the following for clarity. Note that the toxin production rate depends on the inducer concentrations, and thus we have a value of $a$ for each inducer concentration used. The initial condition for the relative frequencies of the two strains in the numerical integrations of *Equation 11* was assumed to be equal to the relative frequencies at the first measurement time point, that is, we assumed that the toxin did not alter the relative frequency of the two strains from inoculation to the first measurement time point. The assumption is justified because both the total cell density and the toxin concentration ($c = 0$ at $t = 0$) are low in the first phases of the dynamics. The initial condition for the total cell density $n_0$, for each choice of $g$ and $K$ in the Markov Chain, was set to $n_0 = n_1 / \left[n_1/K + e^{gt_1}\left(1 - n_1/K\right)\right]$, which is the solution of the logistic equation $dn/dt = gn\left(1 - n/K\right)$ for $n_0$, with $n\left(t_1\right) = n_1$ where $t_1 = 14$ hr is the first measurement time point and $n_1$ was taken from the data. All data were fit simultaneously, because the parameters $g$, $K$, and $b$ appear for all inducer concentration treatments. The best-fit parameters are given in *Table 4*. Note that when comparing the predictions of this model to the data on the antagonistic competition between two killer strains, some of the parameters appear for more than one competition assay. For example, the parameter $a_2^{0uM\ Cu}$ appears in all the competitions between K1 and K2 in which we added only galactose to the medium, and in the competition assay without any inducer.

The parameters of *Equations 7* and *9* were estimated by fitting the following model with Monod growth dynamics (*Monod, 1949*) to the data from competition assays between toxin-producing and sensitive, nonkiller strains:

**Table 5.** Best-fit estimates for the parameters of the model with Monod growth dynamics **Equation 12** fit to the data from competition assays between toxin-producing strains and sensitive, nonkiller ones, via Markov Chain-Monte-Carlo (MCMC).

Concentrations of the inducers galactose (Gal) and copper (Cu) are indicated as superscripts. The parameters $a_1$, $a_2$, and $a_{2b}$ correspond to the rescaled parameter $a' = da$ in **Equation 12** for the strains K1, K2, and K2$_b$, respectively.

| | | | |
|---|---|---|---|
| $g_{max}$ | 1/hr | $a_1^{360\mu M\ Gal}$ | $1.76 \cdot 10^{-9}$ mL/cell/hr$^2$ |
| $Y$ | $1.3 \cdot 10^{10}$ cells/(g glucose) | $a_2^{0\mu M\ Cu}$ | $5.09 \cdot 10^{-10}$ mL/cell/h$^2$ |
| $b$ | $3.2 \cdot 10^{-9}$ mL/cell/hr$^1$ | $a_2^{12.5\mu M\ Cu}$ | $1.04 \cdot 10^{-9}$ mL/cell/h$^2$ |
| $a_1^{0\mu M\ Gal}$ | Set to 0 mL/cell/hr$^2$ | $a_2^{25\mu M\ Cu}$ | $1.24 \cdot 10^{-9}$ mL/cell/h$^2$ |
| $a_1^{200\mu M\ Gal}$ | $1.65 \cdot 10^{-10}$ mL/cell/hr$^2$ | $a_2^{50\mu M\ Cu}$ | $1.43 \cdot 10^{-9}$ mL/cell/h$^2$ |
| $a_1^{240\mu M\ Gal}$ | $2.45 \cdot 10^{-10}$ mL/cell/hr$^2$ | $a_{2b}^{0\mu M\ Cu}$ | $3.47 \cdot 10^{-10}$ mL/cell/h$^2$ |
| $a_1^{280\mu M\ Gal}$ | $6.11 \cdot 10^{-10}$ mL/cell/hr$^2$ | $a_{2b}^{5\mu M\ Cu}$ | $5.52 \cdot 10^{-10}$ mL/cell/h$^2$ |
| $a_1^{320\mu M\ Gal}$ | $1.18 \cdot 10^{-9}$ mL/cell/hr$^2$ | $a_{2b}^{50\mu M\ Cu}$ | $7.57 \cdot 10^{-10}$ mL/cell/h$^2$ |

$$\begin{cases} \dfrac{dn_K}{dt} = g_{max} n_K \dfrac{c_g}{c_g + K_S} \\[2mm] \dfrac{dn_S}{dt} = g_{max} n_S \dfrac{c_g}{c_g + K_S} - d n_S c \\[2mm] \dfrac{dc}{dt} = a n_K \dfrac{c_g}{c_g + K_S} - b \left( n_K + n_S \right) c \\[2mm] \dfrac{dc_g}{dt} = -\dfrac{g_{max}}{Y} \left( n_K + n_S \right) \dfrac{c_g}{c_g + K_S} \end{cases} \tag{12}$$

The toxin concentration and toxin production rate were rescaled as described previously. The initial concentration of glucose was set to the experimental value $c_{g0} = 0.02$ g/mL and the value for the Monod constant $K_S = 2 \cdot 10^{-5}$ g glucose/mL (0.11 mM) was taken from the literature (**Postma et al., 1989**). The value of $K_S$ does not impact the results significantly within a biologically reasonable range, given that it only affects the later phases of the dynamics when glucose is depleted. The initial condition for the relative frequencies of the two strains in the numerical integrations of **Equation 12** was assumed to be equal to the relative frequencies at the first measurement time point. The initial condition for the total cell density $n_0$, for each choice of the other parameters in the Markov Chain, was set equal to the solution of the growth equation without antagonistic interactions $\frac{dn}{dt} = g_{max} n \frac{c_g}{c_g + K_S}$ with $\frac{dc_g}{dt} = \frac{-g_{max}}{Y} n \frac{c_g}{c_g + K_S}$ for $n_0$ (here, $n = n_K + n_S$), which is also the solution of $n_0 - n\left(t\right) + Y K_S \ln \left( 1 - \frac{n(t) - n_0}{Y c_{g0}} \right) = -g_{max} t$ for $n_0$ (we used the fact that in this simplified model $n - n_0 = Y(c_{g0} - c)$), which we computed numerically using $t = t_1 = 14$ hr and $n\left(t\right) = n_1$ taken from the data (first measurement time point). All data were fit

**Table 6.** Values used in the numerical integrations of **Equations 6–9** for the diffusion coefficients and Monod's constant for growth of *Saccharomyces cerevisiae* on glucose.

The value of $K_S$ was taken from **Postma et al., 1989**. As an estimate for the K1 and K2 toxins diffusion coefficient, we took a typical value for proteins of size similar to the K1 and K2 toxins (**Magliani et al., 1997**) diffusing in agar gels at 25°C (**Pluen et al., 1999**). The yeast diffusion coefficient $D_y$ was estimated as discussed in the text. The value for $D_g$ was taken from **Longsworth, 1955**.

| $D_g$ | $D_t$ | $D_y$ | $K_S$ |
|---|---|---|---|
| 0.024 cm$^2$/hr | $3 \cdot 10^{-3}$ cm$^2$/hr | $3 \cdot 10^{-7}$ cm$^2$/hr | $2 \cdot 10^{-5}$ g glucose/mL |

simultaneously using MCMC, because the parameters $g_{max}$, $Y$, and $b$ appear for all inducer concentration treatments. The best-fit parameters are given in **Table 5**. Also for this model, when comparing the prediction of the model to the data on the antagonistic competition between two killer strains, some of the $a$ parameters appear for more than one competition assay.

Confidence intervals for the model predictions in **Figure 3** were computed by plotting the model predictions using the interaction coefficients $r_1 \pm \sigma_1$ and $r_2 \pm \sigma_2$, where $r_1$ and $r_2$ are the interaction coefficients best-fit estimates and $\sigma_1$ and $\sigma_2$ are the standard deviations reported in **Table 3**.

The diffusion coefficient of glucose in **Equations 6–9** was taken from the literature, whereas the toxin diffusion coefficients were estimated based on experimental values for proteins of similar sizes (**Magliani et al., 1997**) diffusing in agar gels (**Pluen et al., 1999**). Their values are reported in **Table 6**. The yeast diffusion coefficient was estimated as follows. The local movement of cells in our system is not predominantly due to Brownian motion, but rather to the forces that dividing cells exert on each other during growth. Thus, we estimated the yeast diffusion coefficient as $D_y = d^2 g_{max}$, where the typical yeast cell diameter $d \approx 10$ μm (estimated by measuring the mean cell diameter in liquid cultures of K1 and K2) and $g_{max}$ appear for dimensional reasons. Because cells are not locally in isolation and multiple cells are typically dividing and pushing each other at the same time, the effective value of $D_y$ may be larger in the experiments. Other factors that can contribute to increasing the effective diffusion coefficient are the collisions of local clusters of cells that are transferred from old to new plates with by replica plating, and the replica plating itself might also contribute given that the agar surface is pressed onto a microfiber cloth twice to transfer cells to a fresh plate. Increasing the value of $D_y$ in the simulations increases the critical inoculum size. For example, with $D_y = 3 \cdot 10^{-6}$ cm$^2$/hr (10 times larger than the value used here), the critical inoculum size is about 20 mm$^2$. In **Figure 7—figure supplement 3** we used numerical simulations to investigate the dependence of the critical inoculum size on the toxin production rate of the invader and on the toxin diffusion rate in the model. We found that the critical inoculum size in our model is proportional to $\sqrt{D_t}$, as shown by the fact that simulation data collapse onto a single curve when dividing the critical radius by this factor (**Figure 7—figure supplement 3B-C**). Additionally, simulation data suggests that for large $a_1 - a_2$, the critical radius scales as $1/\sqrt{a_1 - a_2}$ (**Figure 7—figure supplement 3C**). In **Figure 7—figure supplement 4**, we also show that the width of the halo varies with the toxin production rates of the two strains and with the diffusion coefficients of glucose and of the toxins.

## Numerical integration of the spatially structured model

We numerically integrated **Equation 7** in polar coordinates and **Equation 8–9** in cylindrical coordinates, using the forward Euler method and assuming symmetry in the azimuthal coordinate. The integration steps in the radial, altitudinal, and temporal directions were set to $dr = 25$ μm, $dz = 50$ μm, and $dt = 10^{-4}$ hr. The parameters of the model were set to the values reported in **Table 5** and **Table 6** (with $a_1 = a_1^{360uM\ Gal}$), except for $a_2$ which was set to $0.85 \cdot 10^{-9}$ mL/cell/hr$^2$ to account for the increased killer strength of copper-induced K2 killer strains on agar plates, compared to liquid cultures. This value was calculated as follows. Upon interpolating between the last two data points in the lower-right panel of **Figure 3B** to estimate the value of $f_{eq}$ for the competition of strains K1 and K2 on agar plates, and assuming that the toxin production rate for strain K1 is unchanged with respect to liquid cultures, we found the estimate $a_{2,plates}^{0uM\ Cu} = 1.2 \cdot 10^{-9}$ mL/cell/hr$^2$ for strain K2 grown on plates with 0 μM copper and 350 μM galactose. Given that K2$_b$ was a weaker killer than K2 by a factor $a_{2b}^{0uM\ Cu}$ / $a_2^{0uM\ Cu} \approx 0.7$ in liquid media with 0 μM copper (**Table 5**), we set $a_{2b,plates}^{0uM\ Cu} = 0.7 a_{2,plates}^{0uM\ Cu} = 0.85 \cdot 10^{-9}$ mL/cell/hr$^2$. The initial condition for the radial density profiles of the two strains was set to reproduce the experiments as closely as possible. To this end, we first reproduced experimentally the initial conditions of the experiments of **Figure 5** by inoculating 22 droplets of different volumes from an overnight culture of strain K1 on a lawn of strain K2$_b$ using the same protocol as for the experiments of **Figure 5**, and droplet volumes ranging from 0.5 to 3 μL. We imaged the spatial distribution of cells of the two types with a fluorescence stereomicroscope at the highest magnification and analyzed the images using custom scripts written in Fiji and Mathematica to reconstruct the radial distribution of the two strains. We measured the cell densities of the two strains in the interior of the droplets, in the coffee stain rings, and outside the droplets, as well as the droplets' radii and the width of the coffee stain rings. We integrated **Equations 7–9** for each of these droplets separately (different curves in **Figure 7**). We have also integrated the model numerically starting from more idealized

initial conditions in which strain K1 occupied the region $r \leq r_0$ and the strain K2$_b$ occupied the region $r \geq r_0$ (i.e., the cell density of K2$_b$ was set to zero for $r \leq r_0$), both with a density equal to the average experimental density of the experimental lawn of strain K2$_b$. The dynamics of this simpler, but less accurate, simulation (*Figure 7—figure supplement 1*) is similar to the one shown in *Figure 7A–C*, with a critical inoculum size of about 14 mm$^2$. In the numerical integrations of the spatial model, we simulated the replica plating transfer as a dilution that preserved the relative density of the two strains at each point in space, reducing their absolute density by a factor 10$^4$ and resetting the concentration of the two toxins in the agar to zero. Varying the dilution factor in the range $[10^3 - 10^5]$ has almost no discernible effect on the model's output. Numerical integrations were performed using custom Matlab scripts.

## Acknowledgements

We thank Manuel Ramírez for sending us reference K1, K2, and K⁻ strains, Elena Servienė for sending us the plasmid with the K2 DNA copy, and Manfred Schmitt and Frank Breinig for sending us the plasmid YES2.1/V5-HIS-TOPO-K1 pptox. We thank the members of the AWM and DRN groups for insightful comments on the research performed. AG thanks Marco Fumasoni for assistance in cloning. We thank Michael Laub and Michael Desai for insightful comments on the manuscript. We thank Matti Gralka and two anonymous referees for their insightful comments. AG was supported by the Swiss National Science Foundation, Projects P2ELP2_168498 and P400PB_180823, and work on this project was supported by the Human Frontier Science Program Grant RGP0041/2014 (to AWM and DRN) and NSF/Simons Center for Mathematical and Statistical Analysis of Biology at Harvard (#1764269 [NSF] and #594596 [Simons Foundation]). Work by AG and DRN was supported by the National Science Foundation, via Grant DMR1608501 and via the Harvard Materials Science Research and Engineering Center via Grant DMR-2011754.

## Additional information

### Funding

| Funder | Grant reference number | Author |
| --- | --- | --- |
| Swiss National Science Foundation | P2ELP2_168498 | Andrea Giometto |
| Swiss National Science Foundation | P400PB_180823 | Andrea Giometto |
| Human Frontier Science Program | RGP0041/2014 | David R Nelson<br>Andrew W Murray |
| National Science Foundation | 1764269 | Andrew W Murray |
| Simons Foundation | 594596 | Andrew W Murray |
| National Science Foundation | DMR1608501 | David R Nelson |
| Harvard Materials Research Science and Engineering Center | DMR-2011754 | David R Nelson |

The funders had no role in study design, data collection and interpretation, or the decision to submit the work for publication.

### Author contributions

Andrea Giometto, Conceptualization, Data curation, Formal analysis, Funding acquisition, Investigation, Project administration, Software, Writing - original draft, Writing - review and editing; David R Nelson, Andrew W Murray, Conceptualization, Funding acquisition, Project administration, Resources, Supervision, Writing - original draft, Writing - review and editing

### Author ORCIDs
Andrea Giometto [iD] http://orcid.org/0000-0002-0544-6023
Andrew W Murray [iD] http://orcid.org/0000-0002-0868-6604

### Decision letter and Author response
Decision letter https://doi.org/10.7554/eLife.62932.sa1
Author response https://doi.org/10.7554/eLife.62932.sa2

## Additional files

### Supplementary files
• Transparent reporting form

### Data availability
All data are included in the manuscript and supporting files. All source code that generated figures and numerical results has been uploaded on GitHub at the URL: https://github.com/andreagiometto/Giometto_Nelson_Murray_2020 (copy archived at https://archive.softwareheritage.org/swh:1:rev:18a61001d0cfe2edb3abad40f971f81e3fb1be9a). Source data files have been provided for all Figures displaying data.

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
