## [Decision Letter]

**Decision letter after peer review:**

Thank you for submitting your article "Antagonism between killer yeast strains as an experimental model for biological nucleation dynamics" for consideration by *eLife*. Your article has been reviewed by 3 peer reviewers, and the evaluation has been overseen by a Reviewing Editor and Naama Barkai as the Senior Editor. The following individual involved in review of your submission has agreed to reveal their identity: Matti Gralka (Reviewer #1).

The reviewers have discussed the reviews with one another and the Reviewing Editor has drafted this decision to help you prepare a revised submission.

As you will see below, all reviewers found the paper interesting and important. A series of concerns me questions raised by all reviewers concerns the nutrient depletion explanation of the halos. The specific assumption made in the model (and not confirmed by the experiment) may turn out to be inevitable. Nevertheless, please address the concerns regarding the assumptions on which the model based, as well as all other comments, as specified by the detailed reviews below.

*Reviewer #1:*

The authors present a beautifully clean model system to study mutual antagonism in spatial and well-mixed populations which they use to test theoretical predictions from models developed previously (and adapted here). In particular, the authors verify the prediction that in a spatial setting a stronger antagonist can invade a population of a weaker antagonist only if the former is present at a large enough initial population size. The paper is well written and very accessible, and I have only two substantive concerns:

1. Given the focus on minimum population size for invasion in the introduction I expected the authors to present the same verification also for well-mixed populations. While Figure 3B seems to indicate that in well-mixed populations the relative abundance of the stronger antagonist can indeed decrease if its initial abundance is below a threshold value I would have liked to see this shown (or at least stated in the text) more clearly.

2. In the later portions of the manuscript, the authors infer from a variety of models they tested that the halo separating the two genotypes stems from nutrient depletion. To make their conclusions about nutrient depletion more comprehensible it would be helpful to show the results of models of intermediate complexity and their lack of a halo effect. The authors then go on to suggest (l. 495) that resistant mutants can re-invade the invading strains by presumably crossing the halo into the invader's territory; but how do the mutants cross the nutrient-depleted halo?

*Reviewer #2:*

In this manuscript the authors report interesting results on the population dynamics of a yeast population containing two antagonistic strains. The authors designed the yeast strains to produce toxins in variable amounts, which are regulated by inducible promoters. This enables them to systematically study the role of antagonistic interactions under controlled experimental conditions in different environments, including well mixed liquid cultures and agar surfaces. Specifically, they are investigating under which conditions the "stronger" of the two strains is able to invade a population of the "weaker" strain. Their main result is that the invasion requires a threshold population size (nucleation threshold) of the stronger strain. While the existence of such a threshold is less surprising (due to the state-dependence of the competition), the quantification of the effect in well-mixed and spatially extended systems is interesting. In particular, the authors give a fairly quantitative mathematical description of population dynamics using a rate equation approach for the well-mixed system and a reaction-diffusion model for the spatially extended system. This mathematical analysis provided further insights into the nature of competition. This includes that it is possible to describe the well mixed system in the form of a frequency model using interaction parameters determined from the interaction of the toxin-producing strains with sensitive strains. Furthermore, the reaction-diffusion equations, which explicitly consider both the toxin and the nutrients, seem to capture the formation of a depletion zone (halo) between the two antagonistic strains at the start of their interaction.

The paper is well written and interesting to read. In principle I am in favour of recommending the manuscript for publication. However, I have a major point of criticism that the authors should consider before I can make final recommendations.

When investigating the nature and role of the different mutants found in their studies, the authors do not discuss the possibility of changes in growth rates. While the engineered original strains show the same growth rates, this might not be the case for the mutant strains. This could have a major impact on the dynamics, since, for example, the validity of a pure frequency model depends on identical growth rates. I would ask the authors to measure these growth rates in their mutant strains. Furthermore, I think it should be straightforward to test possible effects of a change in growth rates with their reaction-diffusion model.

*Reviewer #3:*

In addition to the technical questions raised above I have a few quibbles about the presentation.

– The authors start using the term "stronger antagonist" already in the abstract without giving it even any definition, so the reader has to guess the meaning the best she/he can. Definition comes on line 118, page 8…

– It would be useful to have a more explicit layout for the paper early on. This reader was wondering about effects of toxin stability and diffusion all through the well-mixed section, without knowing that it is coming later.

Overall, I think the manuscript is suitable for *eLife* after suitable revisions/clarifications.

---

## [Author Response]

As you will see below, all reviewers found the paper interesting and important. A series of concerns me questions raised by all reviewers concerns the nutrient depletion explanation of the halos. The specific assumption made in the model (and not confirmed by the experiment) may turn out to be inevitable. Nevertheless, please address the concerns regarding the assumptions on which the model based, as well as all other comments, as specified by the detailed reviews below.

In response to the reviewers and your suggestion, we have now expanded our discussion of intermediate models and how they fail to reproduce the halo that separates the two competing strains. We do agree that we haven’t directly tested the hypothesis that the formation of the halos is due to the combined effect of the toxin activity *and* of the depletion of nutrients. However, we cannot come up with a simple way of testing whether the toxin activity alone is sufficient to cause the formation of the halo. One could imagine a setup in which nutrients are continuously provided to the agar, but this would cause an indefinite growth of the two populations, which would lead to a very thick lawn on top of the agar, and the physiological state and physical environment of cells within such lawn would be very different from that of our experiment. Most likely, such indefinite growth would lead to the halo being filled with cells, as it is hard to imagine that a ‘canyon’ would persist with no ‘landslides’ in such configuration. Alternatively, one could envision doing an experiment in a sophisticated setup using microfluidic chambers that would only allow a monolayer of cell to grow within them, but the pressure buildup within the chamber would also most likely lead to the halo being filled with cells. In both cases, the physical environment would be very different from the one experienced by the cells in our experiments. For these reasons, we rely on modeling to explain the halo formation dynamics, using a combination of experimentally measured and physically realistic parameters. Incidentally, the increased density of cells at the two sides of the halo (compared to anywhere else on the plates, visible as increased fluorescence intensity in Author response image 1) observed in the experiments is indirect evidence for the diffusion of nutrients away from the halo, supporting the growth of cells at the two sides of it.

**Author response image 1. sa2fig1:** An increased fluorescent intensity was often observed at the two sides of the halo, corresponding to a thicker layer of cells in those regions compared to anywhere else on the plate. Modeling suggests that such thicker layer of cells is due to nutrient diffusion away from the halo.

We have now included another line of evidence for the diffusion of nutrients away from the halo region, as described in the following paragraph which was added to the manuscript: (lines 507-514)

“If the halo was caused by the presence of the toxins alone, and not by the combined effect of the toxins and the diffusion of nutrients away from the agar underneath the halo, one would expect that inhibition of the toxin would allow cells to re-invade the halo region. To test this, we experimentally verified that no further growth in the halo region is observed after transferring populations that competed for 48h at 25^o^C to 32^o^C for further 48h (Figure 5 – supplement 3), a temperature at which both the K1 and K2 toxins are unstable and fail to inhibit the growth of susceptible strains (Marquina, Santos, and Peinado, 2002, Lukša, Serva, and Serviene, 2016, Figure – supplement 4).”

Although we failed to comment on this in the previous version of the manuscript, there is a simple explanation for the failure of the model with logistic growth term to produce the halo, and we have now added a discussion on this point in the revised version of the manuscript (lines 837-849). We thank you and the reviewers for pointing out the missing explanation. The rationale is as follows.

The logistic growth term assumes that every cm^2^ on the agar can support a given number of cells. In such a model, nutrients located in a given region of space cannot diffuse to nearby regions and thus can only support cells locally. With toxin production rates representative of our experiments, the toxin produced by an antagonist strain is not sufficient to completely halt the growth of the other antagonist. In the model with logistic growth, therefore, the two populations are able to grow at the interface between the two antagonist strains, even if at a slower pace compared to other regions of space, eventually filling the halo region with cells in the limit of large times. When nutrients can diffuse, however, nutrients move to other regions of space before cells at the interface between the two antagonists are able to grow, leading to the depletion region that we referred to as the ‘halo’.

Reviewer #1:The authors present a beautifully clean model system to study mutual antagonism in spatial and well-mixed populations which they use to test theoretical predictions from models developed previously (and adapted here). In particular, the authors verify the prediction that in a spatial setting a stronger antagonist can invade a population of a weaker antagonist only if the former is present at a large enough initial population size. The paper is well written and very accessible, and I have only two substantive concerns:1. Given the focus on minimum population size for invasion in the introduction I expected the authors to present the same verification also for well-mixed populations. While Figure 3B seems to indicate that in well-mixed populations the relative abundance of the stronger antagonist can indeed decrease if its initial abundance is below a threshold value I would have liked to see this shown (or at least stated in the text) more clearly.

Thank you for this comment, we have now highlighted this point more clearly in the main text at lines 192-198:

The equilibrium frequencyf _eq_ thus represents a critical inoculum size below which the invasion of a stronger antagonist is predicted to fail in well-mixed settings. Note that this particular “size” relates to an inoculum *concentration* rather than the actual physical size discussed later in this paper for spatially structured communities on surfaces. Nevertheless, when number fluctuations are included in the dynamics, there is an interesting analogy with escape over a barrier problems in statistical mechanics (Chotibut and Nelson, 2015).”

and at lines 248-251:

“Overall, the experimental results from well-mixed experiments (Figure 3) confirm the theoretical prediction that a critical starting frequency, the equilibrium frequency _feq_, is required for a stronger antagonist to invade a resident, antagonist population.”

2. In the later portions of the manuscript, the authors infer from a variety of models they tested that the halo separating the two genotypes stems from nutrient depletion. To make their conclusions about nutrient depletion more comprehensible it would be helpful to show the results of models of intermediate complexity and their lack of a halo effect. The authors then go on to suggest (l. 495) that resistant mutants can re-invade the invading strains by presumably crossing the halo into the invader's territory; but how do the mutants cross the nutrient-depleted halo?

Thank you for this comment, which helped us realize that we hadn’t given sufficient explanation for the failure of the logistic growth model to reproduce the formation of the halo. We have now included the following discussion at lines 836-849 explaining why the models with logistic growth fail to reproduce the formation of the halo:

“The failure of such model to reproduce the halo can be explained as follows. The logistic growth term in Equations 6 assumes that every cm^2^ on the agar can support K_spatial_ cells. In such a model, nutrients located in a given region of space cannot diffuse to nearby regions and thus can only support the growth of cells locally. With toxin production rates representative of our experiments, the toxin produced by an antagonist strain is not sufficient to completely halt the growth of the other antagonist, as shown by the fact that the absolute number of cells of both antagonists grew in all our well-mixed competition experiments, even if the relative frequency of one of the strains declined with time. In the model with logistic growth, the two populations are thus able to grow at the interface between the two antagonist strains, even if at a slower pace compared to other regions of space, eventually almost completely filling the halo region with cells (Figure 7 —figure supplement 1A). When nutrients can diffuse, however, nutrients move to other regions of space before cells at the interface between the two antagonists are able to grow, leading to the depletion region that we referred to as the ‘halo’.”

We have also now included in Figure 7 – supplement 1 plots of numerical simulations of the intermediate models, showing that they fail to reproduce the formation of the halo.

Concerning the ability of the mutants to cross the nutrient-depleted halo, it is important to note that, right after a dilution, nutrients are abundant everywhere on the plate, including in the area corresponding to the halo. In the absence of toxin-resistant mutations, the growth rate of cells is reduced in that area due to the presence of the toxin, so nutrients that are below the area corresponding to the halo can diffuse away from that region of space and can be taken up by nearby cells at the edge of the resident and invader populations. In the presence of a resistant mutant, however, the growth rate of resistant cells at the edge of the halo is not reduced by the presence of the toxin, and thus these cells are able to take up nutrients before they diffuse away from the halo. This early access to nutrients allows these cells to cross the halo. We have now added a sentence to clarify this point at lines 550-554:

;We believe that resistant cells were able to cross the nutrient-depleted region of the halo because, right after the populations were diluted by replica plating, resistant cells could grow and divide despite the presence of the invader’s strain toxin, and they could thus take up nutrients located in that region of space before those nutrients diffused away.”

Reviewer #2:In this manuscript the authors report interesting results on the population dynamics of a yeast population containing two antagonistic strains. The authors designed the yeast strains to produce toxins in variable amounts, which are regulated by inducible promoters. This enables them to systematically study the role of antagonistic interactions under controlled experimental conditions in different environments, including well mixed liquid cultures and agar surfaces. Specifically, they are investigating under which conditions the "stronger" of the two strains is able to invade a population of the "weaker" strain. Their main result is that the invasion requires a threshold population size (nucleation threshold) of the stronger strain. While the existence of such a threshold is less surprising (due to the state-dependence of the competition), the quantification of the effect in well-mixed and spatially extended systems is interesting. In particular, the authors give a fairly quantitative mathematical description of population dynamics using a rate equation approach for the well-mixed system and a reaction-diffusion model for the spatially extended system. This mathematical analysis provided further insights into the nature of competition. This includes that it is possible to describe the well mixed system in the form of a frequency model using interaction parameters determined from the interaction of the toxin-producing strains with sensitive strains. Furthermore, the reaction-diffusion equations, which explicitly consider both the toxin and the nutrients, seem to capture the formation of a depletion zone (halo) between the two antagonistic strains at the start of their interaction.The paper is well written and interesting to read. In principle I am in favour of recommending the manuscript for publication. However, I have a major point of criticism that the authors should consider before I can make final recommendations.When investigating the nature and role of the different mutants found in their studies, the authors do not discuss the possibility of changes in growth rates. While the engineered original strains show the same growth rates, this might not be the case for the mutant strains. This could have a major impact on the dynamics, since, for example, the validity of a pure frequency model depends on identical growth rates. I would ask the authors to measure these growth rates in their mutant strains. Furthermore, I think it should be straightforward to test possible effects of a change in growth rates with their reaction-diffusion model.

Thank you for this interesting observation. We have now measured the growth rates of mutants in two ways:

1. By spreading single cells of each mutant and ancestor cells on a YPD agar plate identical to the ones used in the spatial experiments (e.g., the experiments whose corresponding data are shown in Figures 5 and 6), with no inducers added. Droplets of each strain/mutant were deposited on the plate on a grid at distances of 1 cm from each other and the location of each strain/mutant within the grid was randomized. Cells on the plate were imaged using an inverted microscope with 50X objective every 20 min for 6 h, during which the plate was kept at 25^o^C using a stage-top incubator. The toxin diffusion coefficient is estimated at 0.003 cm^2^/h, and thus the production of the K2 toxin by strain K2_b_ and its mutants does not affect cells of the of the K1 strain and its mutants at a distance of 1 cm on the same plate (the average distance traveled by a toxin molecule over such time is ). During the 6 h, cells formed micro-colonies of up to 32 cells, with the exception of a few non-dividing or very slow dividing cells that we excluded from the analysis, but whose growth rates are shown in the Figure 6 – figure supplement 2. T-tests performed between all pairs of strains/mutants gave no statistically significant differences between their growth rates (all corresponding p-values were larger than 0.05).

2. By growing liquid cultures of each mutant and ancestor strain mixed with a strain sensitive to the toxins produced by the former and expressing a fluorescent protein of the opposite color, measuring the relative density of each strain in pairwise competitions, and diluting daily. These competition assays performed over multiple days and generations can detect smaller fitness differences than the experiments reported above, by measuring the relative frequency of the two strains at different times over multiple generations. Mixed competition assays performed with strains carrying the K1 killer toxin gene induced by the promoter *P_GAL1_* (strains K1, K1^s^ and K1^o^) versus strain S2 can be used to directly detect differences in reproductive fitness, because no toxin is produced in the absence of inducer (galactose) and thus the relative frequency of the two strains varies due to differences in reproductive fitness alone. Mixed competition assays between strains carrying the K2 killer toxin gene induced by the promoter *P_CUP1_* (K2_b_, K2_b_^r^ and K2_b_^o^) versus strain S1, instead, can only detect the joint effect of differences in reproductive fitness *and* killer activity, given that the promoter *P_CUP1_* has non-zero expression even in the absence of its inducer (copper). T-tests performed between all pairs of competitions between killer strains/mutants of the same type (i.e., expressing either the K1 or K2 toxin) and the corresponding sensitive strains (Figure 6—figure supplement 3) give no statistically significant differences between the rates at which the relative frequency of killer strains in competition assays vary with time (all corresponding *p*-values are larger than 0.05). Although the curves on the left plot are not linear, we note that the curves for strains K2_b_, K2_b_^o^ and K2_b_^s^ overlap. The curve for the resistant strain K2_b_^r^ does not overlap with the others, which may be attributable to a difference in the initial relative frequency of K2_b_^r^ and the sensitive strain S1.

Taken together, these results show that no changes in growth rates are detectable between the strains and mutants isolated at the end of the experiment, and between such strains and their ancestors. We have now included these plots in the revised version of the manuscript as Figure 6 —figure supplements 2 and 3. The fact that no differences in growth rate could be detected between any pair of strains is now mentioned in the main text at lines 439-442:

“Figure 6 —figure supplements 2 and 3 reveal that we could not detect any differences in growth rate between any pairs of strains, ruling out the possibility that the altered outcomes of competition observed in Figure 6 could be due to changes in cell division times during the experiment.”

Reviewer #3:In addition to the technical questions raised above I have a few quibbles about the presentation.– The authors start using the term "stronger antagonist" already in the abstract without giving it even any definition, so the reader has to guess the meaning the best she/he can. Definition comes on line 118, page 8…

The definition has now been anticipated to lines 61-63, as there was no space to define it in the abstract:

“We refer to the strain that survives in a 1:1, well-mixed culture as the stronger antagonist and the one that goes extinct as the weaker antagonist. Models based on generalizations of the Lotka-Volterra equations (Lavrentovich and Nelson, 2019, Tanaka, Stone, and Nelson, 2017) predict that being a stronger antagonist (i.e., leading a weaker antagonist to extinction when starting at equal relative abundances) is a necessary, but not a sufficient condition for an invading strain to replace a resident, antagonist population”

It is of course difficult to give a precise definition before having introduced the details of the experimental system, hopefully the adjectives ‘strong’ and ‘weak’ in the abstract make at least intuitive sense.

– It would be useful to have a more explicit layout for the paper early on. This reader was wondering about effects of toxin stability and diffusion all through the well-mixed section, without knowing that it is coming later.

We have now added an explicit layout at lines 98-105. Together with the paragraph at lines 8597 and Figure 1, the layout of the paper should now be clearer.